# Experimentally induced active and quiet sleep engage non-overlapping transcriptional programs in *Drosophila*

Niki Anthoney[1], Lucy Tainton-Heap[1], Hang Luong[2], Eleni Notaras[1], Amber B Kewin[1], Qiongyi Zhao[1], Trent Perry[2], Philip Batterham[2], Paul J Shaw[3], Bruno van Swinderen[1]*

[1]Queensland Brain Institute, The University of Queensland, Brisbane, Australia; [2]School of BioSciences, The University of Melbourne, Melbourne, Australia; [3]Department of Neuroscience, School of Medicine, Washington University in St. Louis, St Louis, United States

**Abstract** Sleep in mammals can be broadly classified into two different physiological categories: rapid eye movement (REM) sleep and slow-wave sleep (SWS), and accordingly REM and SWS are thought to achieve a different set of functions. The fruit fly *Drosophila melanogaster* is increasingly being used as a model to understand sleep functions, although it remains unclear if the fly brain also engages in different kinds of sleep as well. Here, we compare two commonly used approaches for studying sleep experimentally in *Drosophila*: optogenetic activation of sleep-promoting neurons and provision of a sleep-promoting drug, gaboxadol. We find that these different sleep-induction methods have similar effects on increasing sleep duration, but divergent effects on brain activity. Transcriptomic analysis reveals that drug-induced deep sleep ('quiet' sleep) mostly downregulates metabolism genes, whereas optogenetic 'active' sleep upregulates a wide range of genes relevant to normal waking functions. This suggests that optogenetics and pharmacological induction of sleep in *Drosophila* promote different features of sleep, which engage different sets of genes to achieve their respective functions.

*For correspondence:
b.vanswinderen@uq.edu.au

Competing interest: The authors declare that no competing interests exist.

## eLife assessment

*Drosophila* is a powerful model organism for understanding the molecular and neural regulation of sleep. However, methodological limitations exist that would appear to limit the relevance of work done in the fly to our understanding of mammalian sleep. In this **important** work, the authors provide physiological, behavioral, and molecular evidence for the existence of two potential sleep stages in *Drosophila*. The experiments are generally well conducted and the authors' interpretations of their results are **solid** overall. Although technically innovative and conceptually provocative, there are aspects of the approaches used and results obtained that leave the central conclusions open to interpretation.

## Introduction

There is increasing evidence that sleep is a complex phenomenon in most animals, comprising of distinct stages that are characterized by dramatically different physiological processes and brain activity signatures (*Jaggard et al., 2021*; *Van De Poll and van Swinderen, 2021*). This suggests that different sleep stages, such as rapid eye movement (REM) and slow-wave sleep (SWS) in humans and other mammals (*Dijk et al., 1990*), are accomplishing distinct functions that are nevertheless collectively

important for adaptive behavior and survival (*Tononi and Cirelli, 2014*). While REM and SWS appear to be restricted to a subset of vertebrates (e.g., mammals, birds, and possibly some reptiles; *Siegel, 2011*; *Shein-Idelson et al., 2016*; *Rattenborg et al., 2019*), a broader range of animals, including invertebrates, demonstrate evidence of 'active' vs. 'quiet' sleep (*Jaggard et al., 2021*; *Van De Poll and van Swinderen, 2021*; *Libourel and Herrel, 2016*). Whether active and quiet sleep represent evolutionary antecedents of REM and SWS, respectively, remains speculative (*Jaggard et al., 2021*; *Van De Poll and van Swinderen, 2021*). However, during active sleep, although animals are less responsive, brain recordings reveal a level of neural activity that is similar to wakefulness, in contrast to quiet sleep, which is characterized by significantly decreased neural activity in invertebrates (*Yap et al., 2017*; *Tainton-Heap et al., 2021*) as well as certain fish (*Leung et al., 2019*), mollusks (*Iglesias et al., 2019*), and reptiles (*Shein-Idelson et al., 2016*).

Although it is likely that even insects such as fruit flies and honeybees sleep in distinct stages (*Sauer et al., 2003*; *van Alphen et al., 2013*), sleep studies using the genetic model *Drosophila melanogaster* still mostly measure sleep as a single phenomenon, defined by 5 min (or more) of inactivity (*Shaw et al., 2000*; *Hendricks et al., 2000*). As sleep studies increasingly employ *Drosophila* to investigate molecular and cellular processes underpinning potential sleep functions, this simplified approach to measuring sleep in flies carries the risk of overlooking different functions accomplished by distinct kinds of sleep. Sleep physiology and functions are increasingly being addressed in the fly model by imposing experimentally controlled sleep regimes, either pharmacologically or via transient control of sleep-promoting circuits by using opto- or thermogenetic tools (*Shafer and Keene, 2021*). Yet, there is little knowledge available on whether these different approaches are producing qualitatively similar sleep. For example, sleep can be induced genetically in flies by activating sleep-promoting neurons in the central complex (CX) – a part of the insect brain that has been found to be involved in multimodal sensory processing (*Wolff and Rubin, 2018*). In particular, the dorsal fan-shaped body (dFB) of the CX has been proposed to serve as a discharge circuit for the insect's sleep homeostat, whereby increased sleep pressure (e.g., due to extended wakefulness) alters the physiological properties of dFB neurons, causing them to fire more readily and thereby promote decreased behavioral responsiveness (*Troup et al., 2018*) and thus sleep (*Donlea et al., 2011*; *Donlea et al., 2014*; *Donlea et al., 2018*). Crucially, activation of sleep-promoting circuits including the dFB has been shown to be sleep-restorative (*Tainton-Heap et al., 2021*; *Dissel et al., 2015*), but confusingly, brain recordings during optogenetically induced sleep, via electrophysiology, as well as whole-brain calcium imaging techniques, reveal wake-like levels of brain activity (*Yap et al., 2017*; *Tainton-Heap et al., 2021*). This suggests that some approaches to optogenetically induced sleep might be promoting a form of sleep akin to the 'active' sleep stage detected during spontaneous sleep (*Yap et al., 2017*; *Tainton-Heap et al., 2021*).

An alternate way to induce sleep in *Drosophila* is by feeding flies drugs that have been designed to treat insomnia in humans, such as the GABA-agonist 4,5,6,7-tetrahyrdoisoxazolopyridin-3-ol (THIP), also known as gaboxadol (*Harrison, 2007*). Several studies have shown that THIP-induced sleep in flies is also restorative and achieves key functions ranging from memory consolidation to cellular repair and waste clearance (*Dissel et al., 2015*; *Berry et al., 2015*; *Stanhope et al., 2020*; *van Alphen et al., 2021*). This pharmacological approach centered on GABA function has a solid foundation based on better-understood sleep processes: in mammals, many sleep-inducing drugs also target GABA receptors, and this class of drugs tends to promote SWS (*Lundahl et al., 2012*). In contrast, there are no obvious drugs that promote REM sleep, although local infusion of cholinergic agonists (e.g., carbachol) to the brainstem has been shown to induce REM-like states in cats (*Mitler and Dement, 1974*).

In this study, we compare THIP-induced sleep with optogenetically induced sleep in *Drosophila* using behavior, brain activity, and transcriptomics. To ensure the validity of our comparisons, we performed all of our experiments in the same genetic background, employing a canonical Gal4 strain that expresses a transgenic cation channel in a sleep-promoting circuit: R23E10-Gal4>UAS-Chrimson (*Jenett et al., 2012*; *Klapoetke et al., 2014*). When these flies are fed all-trans-retinal (ATR) and then exposed to red light, they are put to sleep optogenetically. When these flies are instead fed THIP, they are put to sleep pharmacologically. By using the same genetic background, we were thus able to contrast the effects of either kind of sleep at the level of behavior, brain activity, and gene expression (*Figure 1*). We questioned how similar either form of induced sleep was.

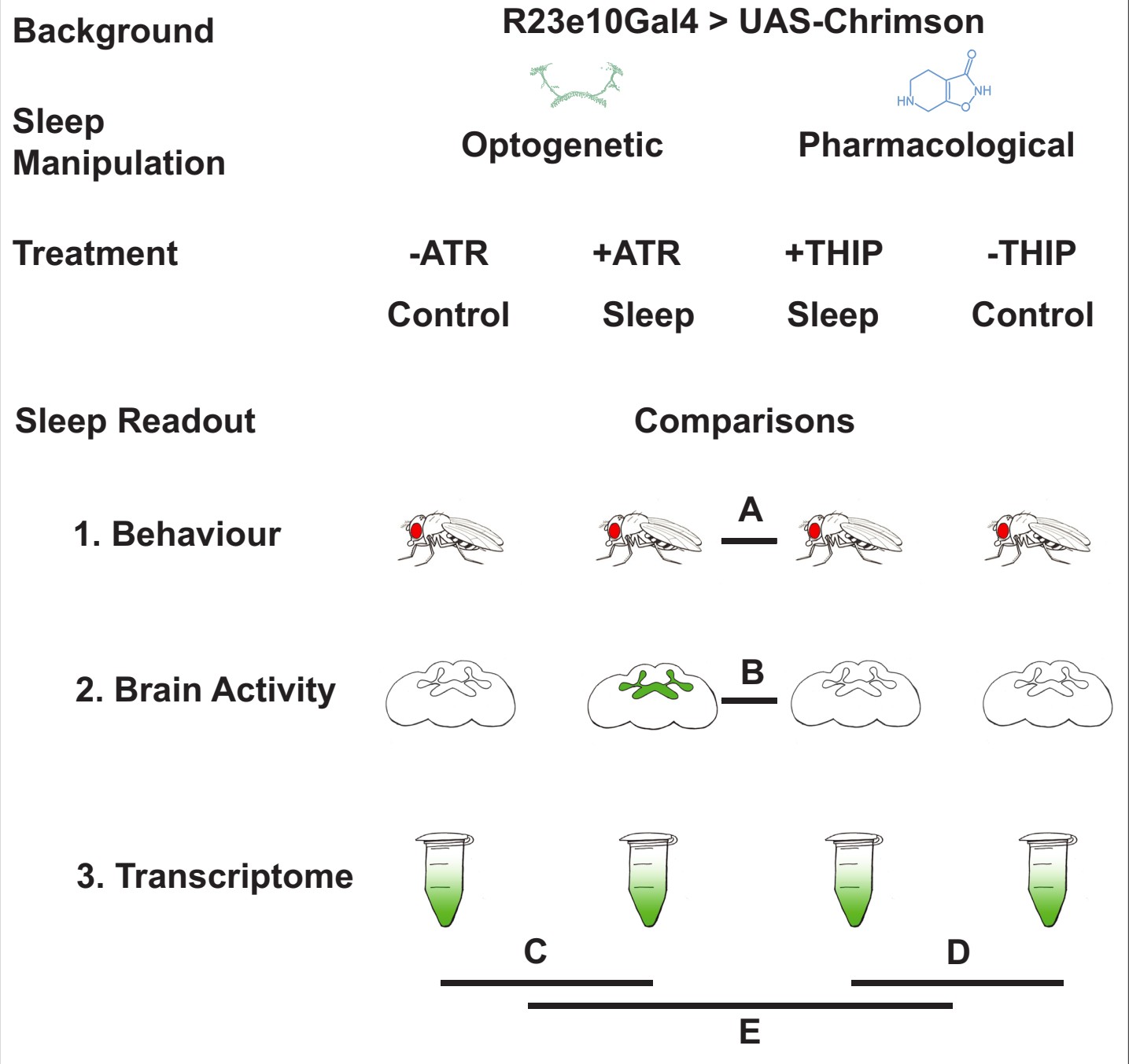

**Figure 1.** Study rationale and design. The same genetic background strain (R23E10;UAS-Chrimson) was used for optogenetic or pharmacologically induced sleep. Flies were fed either all-trans retinal (ATR) or 4,5,6,7-tetrahyrdoisoxazolopyridin-3-ol (THIP) to promote either kind of sleep, which was assessed in three different ways: behavioral analysis, whole brain imaging, and gene expression changes. The comparisons made for each level of analysis are labeled A–E.

## Results

### Prolonged optogenetic- and THIP-induced sleep have near-identical effects on sleep duration

We first compared pharmacological and optogenetic sleep (*Figure 1*) by using the traditional behavioral metrics employed by most *Drosophila* sleep researchers: >5 min inactivity for flies confined in small glass tubes over multiple days and nights (*Shaw et al., 2000*; *Hendricks et al., 2000*). We

found that optogenetic- and THIP-induced sleep yielded almost identical effects on sleep duration, with both significantly increasing total sleep duration for both the day and night, when compared to controls (*Figure 2A–D*, *Supplementary file 1*). An increase in total sleep duration can be due to either an increase in the number of sleep bouts that are occurring (reflective of more fragmented sleep) or an increase in the average duration of individual sleep bouts, which indicates a more consolidated sleep structure (*Zimmerman et al., 2008b*; *Huber et al., 2004*; *Andretic and Shaw, 2005*). To investigate whether both sleep induction methods also had similar effects on sleep architecture, we plotted bout number as a function of bout duration for optogenetic- and THIP-induced sleep for the day and night (*Kirszenblat et al., 2019*). We found that both optogenetic activation and THIP provision produce a similar increase in sleep consolidation during the day (*Figures 2E and F and 3*). During the night, induced sleep effects were also similar, although less clearly different to the spontaneous sleep seen in control flies (*Figures 2G and H and 3*). Interestingly, red light exposure decreased average night bout duration in non-ATR control flies (*Figure 3A–D*), suggesting a light-induced artifact at night. For THIP, we observed an increase in both bout number and duration during the day, and an increase in bout duration during the night (*Figure 3E and F*). Taken together, these results show that prolonged optogenetic activation and THIP provision have similar behavioral effects on induced sleep duration in *Drosophila*, although some differences in night bout architecture were noted (*Figure 3*). Without any further investigations, this might suggest that both sleep induction methods represent similar underlying processes that uniformly increase sleep in flies.

## THIP-induced sleep decreases brain activity and connectivity

The brain presents an obvious place to look for any potential differences between sleep induction methods. In a previous study employing whole-brain calcium imaging in tethered flies, we showed that optogenetic activation of the R23E10 circuit promotes wake-like sleep, with neither neural activity levels nor connectivity metrics changing significantly even after 15 min of optogenetically induced sleep (*Tainton-Heap et al., 2021*). We therefore utilized the same fly strain as in that study (R23E10-Gal4>UAS-Chrimson88-tdTomato;Nsyb-LexA>LexOp-nlsGCaMP6f) to examine the effect of THIP-induced sleep on brain activity (*Figure 4A and B*). Since we were interested in comparing acute sleep induction effects on brain activity (as opposed to prolonged sleep induction effects on behavior, as in *Figures 2 and 3*), we adapted our calcium imaging approach to allow a brief perfusion of THIP directly onto the exposed fly brain (*Figure 4A*, see 'Materials and methods'). As done previously for examining optogenetically induced sleep (*Tainton-Heap et al., 2021*), we examined calcium transients in neural soma scanning across 18 optical slices of the central fly brain (*Figure 4B*, left) and identified regions of interest (ROIs) corresponding to neuronal soma in this volume (*Figure 4B*, right, see 'Materials and methods'). As shown previously (*Tainton-Heap et al., 2021*), optogenetically activating the R23E10 circuit renders flies asleep (albeit twitchy at times) without changing the average level of neural activity measured this way (*Figure 4C*). We have previously shown that even 15 min of R23E10 activation fails to produce a 'quiet' sleep stage that is evident after 5 min in spontaneously sleeping flies (*Tainton-Heap et al., 2021*), suggesting that this manipulation promotes ongoing active sleep until the red light is turned off – although this has not been investigated for longer epochs, for example, the chronic activation experiments described in *Figure 2*. To determine the effect of THIP on neural activity in the exact same strain, we transiently perfused onto the fly brain the minimal THIP dosage required to reliably promote sleep in flies within 5 min (0.2 mg/ml) (*Yap et al., 2017*). In contrast to optogenetic-induced sleep, we observed overall decreased neural activity coincident with the flies falling asleep, and flies remained asleep well after the drug was washed out (*Figure 4D*). To ensure that we were actually putting flies (reversibly) to sleep in this preparation, we probed for behavioral responsiveness by puffing air onto the fly once every minute (50 ms duration, 10 psi) (*Figure 5A*). Since the time when flies fell asleep following 5 min of THIP perfusion could be variable (*Yap et al., 2017*), arousal probing during sleep was only initiated after 5 min of complete quiescence (*Figure 5A*, behavioral responsiveness testing). We observed decreased arousability for flies that had been induced to sleep via THIP perfusion (*Figure 5B*). Drug-induced sleep was, however, reversible, with flies returning to baseline levels of behavioral responsiveness to the air puffs ~20–30 min after sleep initiation. This confirmed that the brief exposure to THIP was indeed putting flies to sleep, with an expected sleep inertia lasting the length of a typical spontaneous sleep bout (*Shaw et al., 2000*; *Hendricks et al., 2000*). This

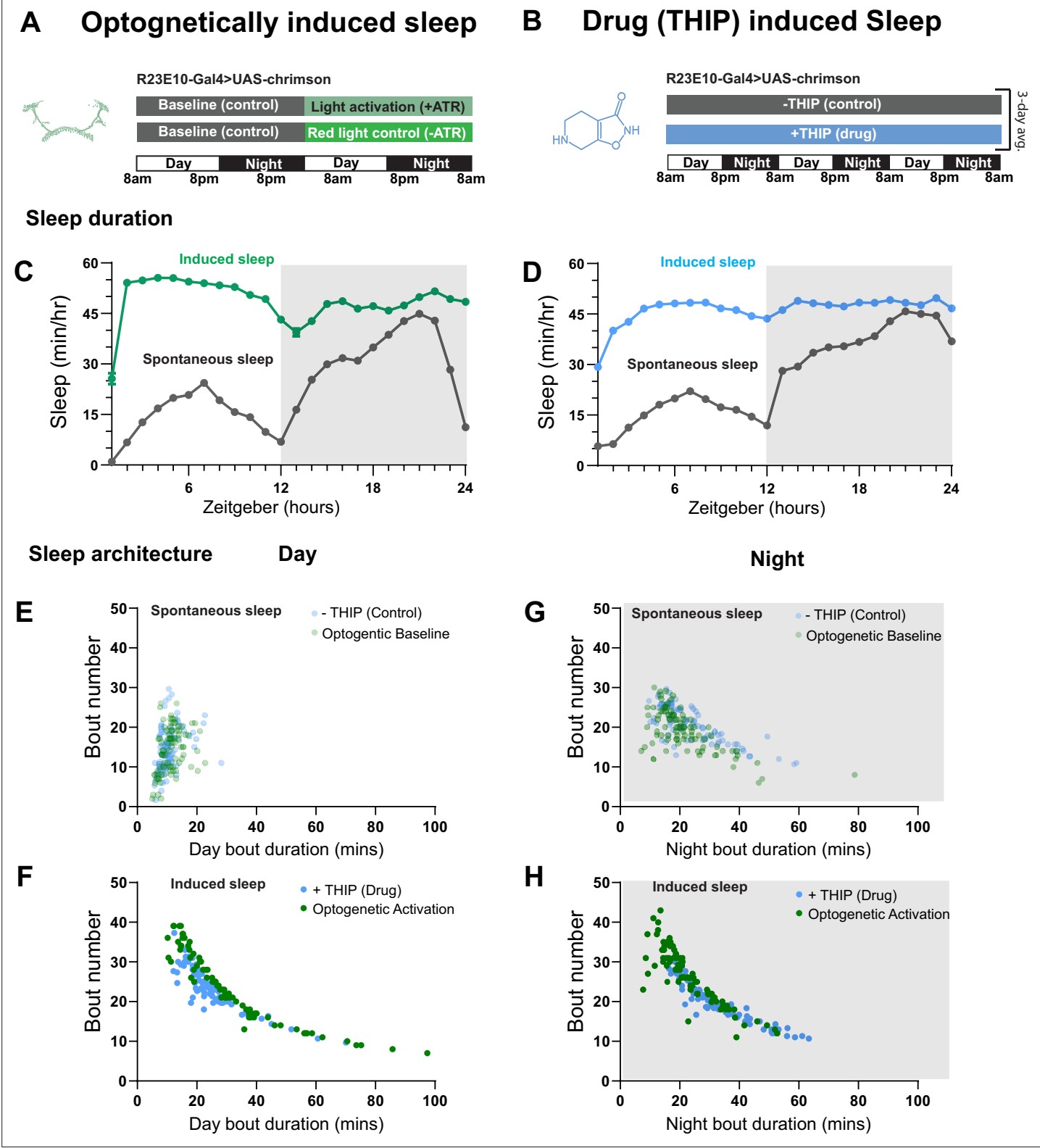

**Figure 2.** Optogenetic- and 4,5,6,7-tetrahyrdoisoxazolopyridin-3-ol (THIP)-induced sleep have similar effects on sleep duration and consolidation. (**A**) Experimental regime for observing the effects of optogenetic activation and THIP provision (**B**). (**C**) Sleep profile across 24 hr in the baseline condition (gray) and optogenetic activation condition (green). (**D**) Three-day average of the 24 hr sleep profile of control (gray) and THIP fed (blue) flies. (**E**) Daytime sleep consolidation scatterplot for optogenetic baseline and THIP control flies. (**F**) Daytime sleep consolidation scatterplot for optogenetic-

*Figure 2 continued on next page*

*Figure 2 continued*

and THIP-induced sleep. (**G**) Night-time sleep consolidation scatterplot for optogenetic baseline and THIP control flies. (**H**) Night-time sleep consolidation scatterplot for optogenetic- and THIP-induced sleep. n = 87 for optogenetic activation across three replicates; n = 88 for –THIP, n = 85 for +THIP, across three replicates. Maximum bout duration possible is 720 min, or 12 hr. See *Supplementary file 1* for statistics.

contrasts with optogenetic sleep induction using the R23E10-Gal4 circuit, which does not appear to show any sleep inertia (*Figure 4C*; *Troup et al., 2023*).

We then examined more closely neural activity in flies that had been put to sleep with THIP. We found that neural activity decreased rapidly within 5 min after sleep onset (*Figure 5C*, +THIP, early). Correlation analysis also revealed a decrease in connectivity among the remaining active neurons (*Figure 5D*, +THIP, early). We also analyzed the next 5 min of sleep and observed similar results

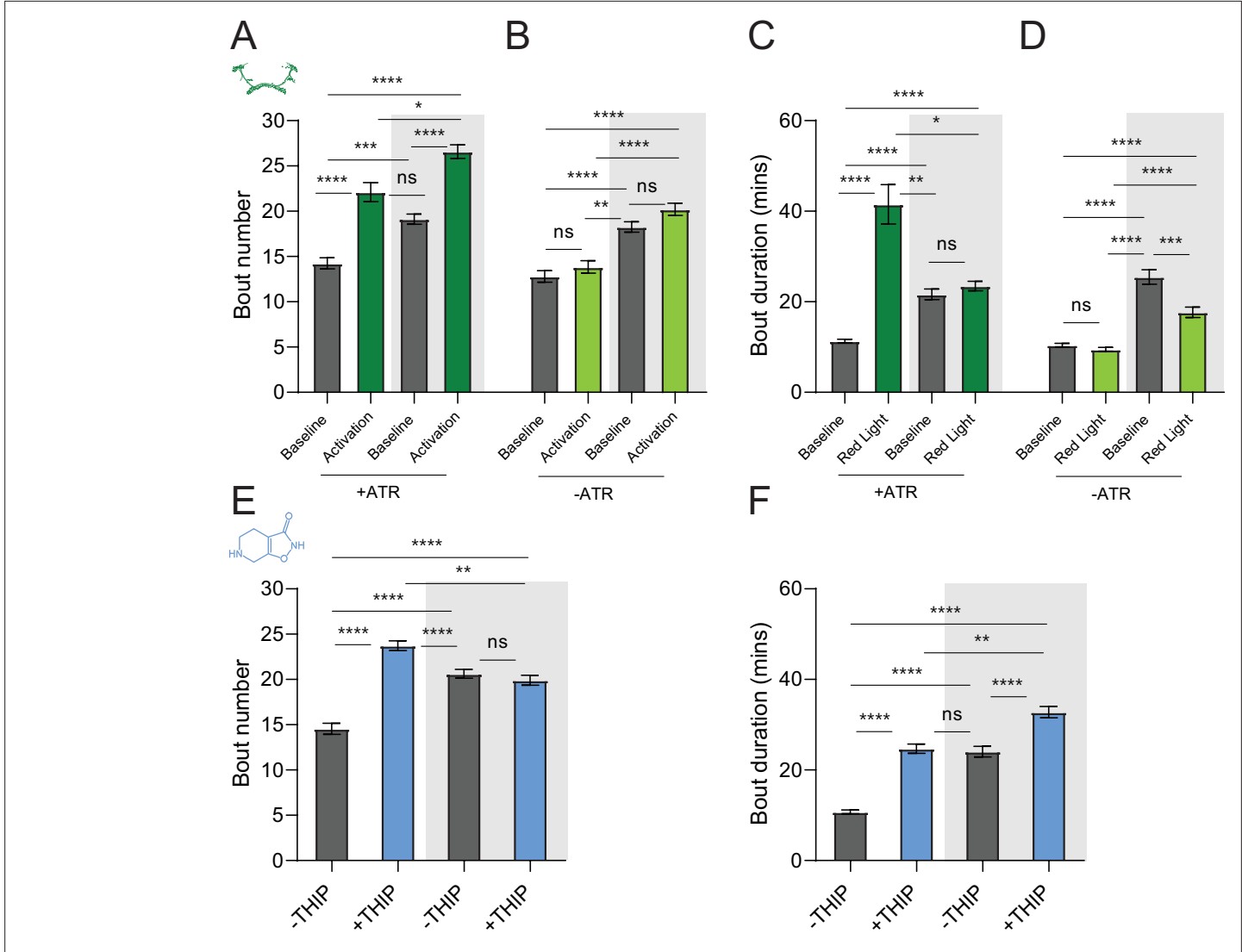

**Figure 3.** Sleep architecture in optogenetic- and 4,5,6,7-tetrahyrdoisoxazolopyridin-3-ol (THIP)-induced sleep. (**A, B**) Average number of sleep bouts in control (gray) and optogenetic activation (green) conditions in the day and night for both +ATR (**A**) and –ATR (**B**) fed flies. Optogenetic-induced sleep results in an increase in the number of sleep bouts both during the day and the night, whereas red light alone has no effect. (**C, D**) Optogenetic activation (green) increases the average sleep duration during the day, but not the night when compared to controls (gray) in +ATR flies (**C**). (**D**) –ATR flies show no difference in mean sleep bout duration during the day, but show a decrease in average bout duration during the night. THIP (blue) increases both the average number of sleep bouts (**E**) and the average duration of sleep bouts (**F**) during the day, but not the night, when compared to controls (gray). Analysis for (**A**) and (**B**) = Kruskal–Wallis test with Dunn's multiple-comparison correction. *p<0.05, ***p<0.001, ****p<0.0001. For (**E**) and (**F**), analysis = ordinary one-way ANOVA with Tukey's correction for multiple comparisons. ***p<0.001, ****p<0.0001.

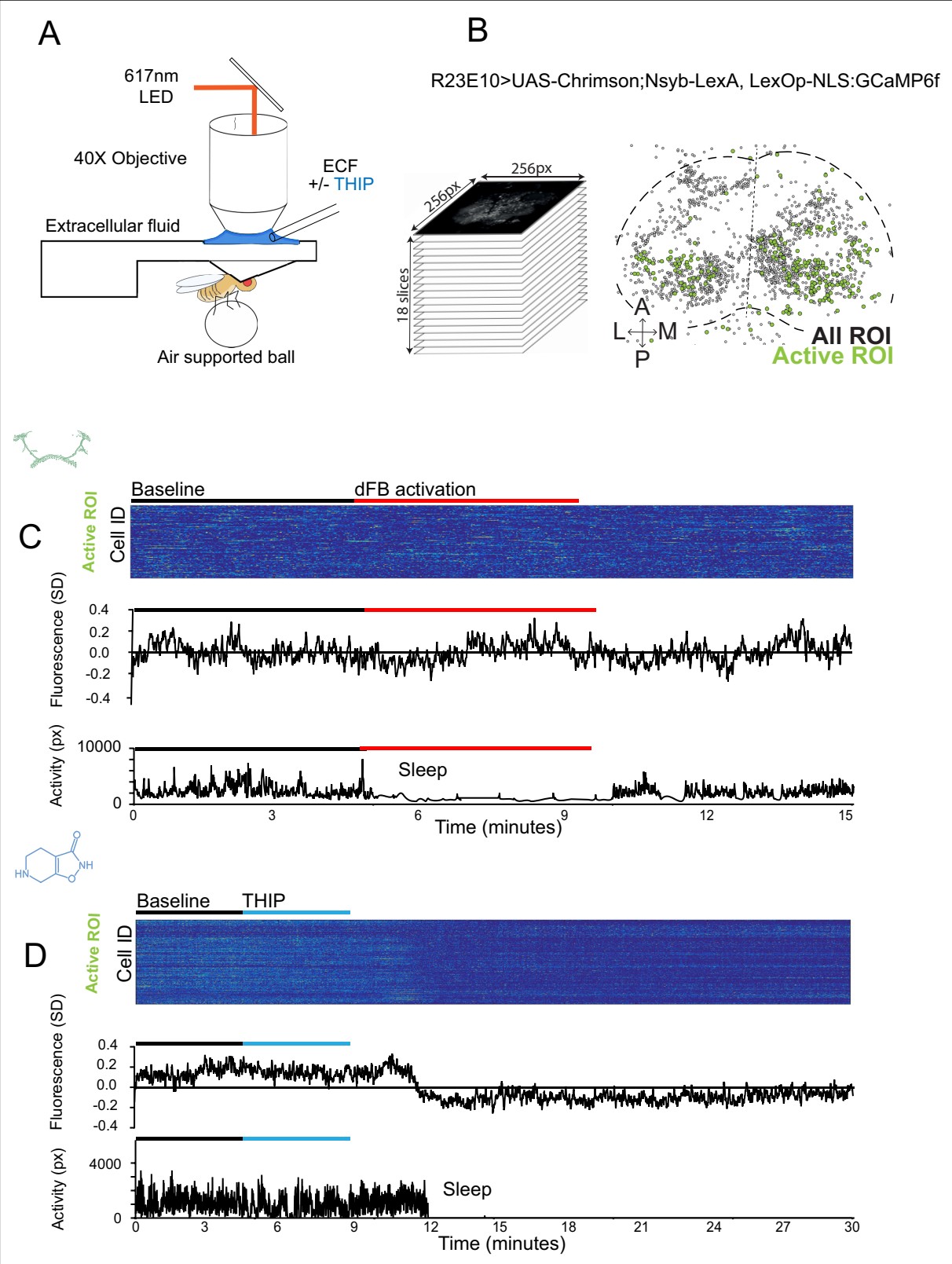

**Figure 4.** Brain imaging during optogenetic- and 4,5,6,7-tetrahyrdoisoxazolopyridin-3-ol (THIP)-induced sleep. (**A**) Flies were mounted onto a custom-built holder that allowed a coronal visualization of the brain through the posterior side of the head. Perfusion of extracellular fluid (ECF) occurred throughout all experiments. A 617 nm LED was delivered to the brain through the imaging objective during optogenetic experiments. During THIP experiments, 4% THIP in ECF was perfused onto the brain through a custom perfusion system. Behavior was recorded as the movement of flies on

*Figure 4 continued on next page*

*Figure 4 continued*

an air-suspended ball. (**B**) Left: imaging was carried out across 18 z-slices, with a z-step of 6 μm. Each z-plane spanned 667 μm × 667 μm, which was captured across 256 × 256 pixels. Right: a collapsed mask from one fly of neurons found to be active (green) in (**C**) alongside all identified regions of interest (ROIs, gray). (**C**) Neural activity in an example fly brain, represented across cells (top) and as the population mean (middle) did not change following optogenetic-induced sleep (bottom). (**D**) Neural activity in an example fly brain, represented across cells (top) and as the population mean (middle) showed an initial high level of activity in the baseline condition, which decreased when the fly fell asleep (bottom) following THIP exposure. The Y-axis scale is standard deviation of the experiment mean, so the baseline is not absolute but rather reflects any difference with the overall experiment mean.

(*Figure 5C and D*, +THIP, mid). All flies eventually woke up from THIP-induced sleep, and brain activity returned to wake levels in three flies that were recorded throughout (*Figure 5C*). These observations suggest that acute THIP exposure is promoting rapid entry into a 'quiet' sleep stage in flies, bypassing the wake-like sleep evident during the first 5 min of spontaneous sleep onset, and closely resembling spontaneous quiet sleep typically seen after 5 minutes of inactivity (*Tainton-Heap et al., 2021*). Importantly, THIP-induced sleep appears to be dissimilar from optogenetically induced sleep in this genotype at the level of neural activity as well as connectivity (*Tainton-Heap et al., 2021*).

In a recent work, we showed that rendering flies unresponsive with a general anesthetic, isoflurane, in surprising contrast to THIP, does not quieten the fly brain (*Troup et al., 2023*). Isoflurane also decreased neural connectivity, as well as the diversity of neural activity across the fly brain, whereas optogenetic-induced sleep did not show any differences in neural activity or ensemble dynamics (*Troup et al., 2023*). Since in our current THIP experiments we were similarly recording from neural soma that we could track through time, we were able to assess the level of overlap between the neurons that remained active during THIP-induced sleep and wakefulness (*Figure 5E and F*). We found that ~30% of active neurons during THIP-induced sleep were also active during wake (*Figure 5F and G*). We next examined whether the same neurons remained active across successive 5 min epochs during THIP-induced sleep compared to wake. We found that there is significantly more overlap between successive 5 min sleep epochs (41%), compared to the waking average (*Figure 5G*), suggesting less neural turnover during THIP-induced sleep than during wake. This contrasts with optogenetic sleep, where the rate of neural turnover was not different from wake (*Troup et al., 2023*). Taken together, our calcium imaging data confirm that pharmacological sleep induction promotes a different kind of sleep than optogenetic sleep induction in the same strain. Henceforth, we call this 'quiet' sleep, in contrast to the 'active' sleep that seems to be engaged by optogenetic activation (*Van De Poll and van Swinderen, 2021*; *Tainton-Heap et al., 2021*). Notably, calcium imaging of spontaneous sleep bouts in *Drosophila* also revealed active and quiet sleep stages (*Tainton-Heap et al., 2021*), suggesting that both of our experimental approaches are physiologically relevant. Whether drug perfusion to the brain is equivalent to feeding is of course less clear. When feeding on 0.1 mg/ml THIP-laced food, flies were continuously exposed to the drug over days, with comparatively less reaching the brain. With perfusion, the brain was directly exposed to 0.2 mg/ml THIP for only 5 min. Interestingly, in both cases this induces daytime sleep bouts, which average around 25 min (*Figures 3F and 5B*), the average duration of a spontaneous night-time sleep bout (*Figure 3F*, *Supplementary file 1*).

## Transcriptional analysis of flies induced to sleep by THIP provision

Our calcium imaging experiments suggest that different biological processes might be engaged by optogenetic-induced sleep compared to THIP-induced sleep. Additionally, we observed neural effects encompassing much of the fly brain (*Figure 5E*) as our recording approach exploited a pan-neural driver. We therefore wondered if either sleep-induction method might lead to differences in gene expression across the whole brain and if these might highlight distinct molecular pathways engaged by either kind of sleep. To address this, we collected brains from flies that had been induced to sleep by either method and compared the resulting transcriptomes with identically handled control animals that had not been induced to sleep by these methods.

To control for genetic background, we again used the same R23E10-Gal4>UAS-Chrimson flies as in our multiday behavioral experiments and fed the flies either THIP or ATR, as in *Figure 2*. We only examined daytime sleep-induction effects for either method as this is when we observed the greatest increase in sleep compared to controls (*Figure 2*), and previous work has shown that daytime sleep induction using either method achieves sleep functions (*Tainton-Heap et al., 2021*; *Dissel et al.,*

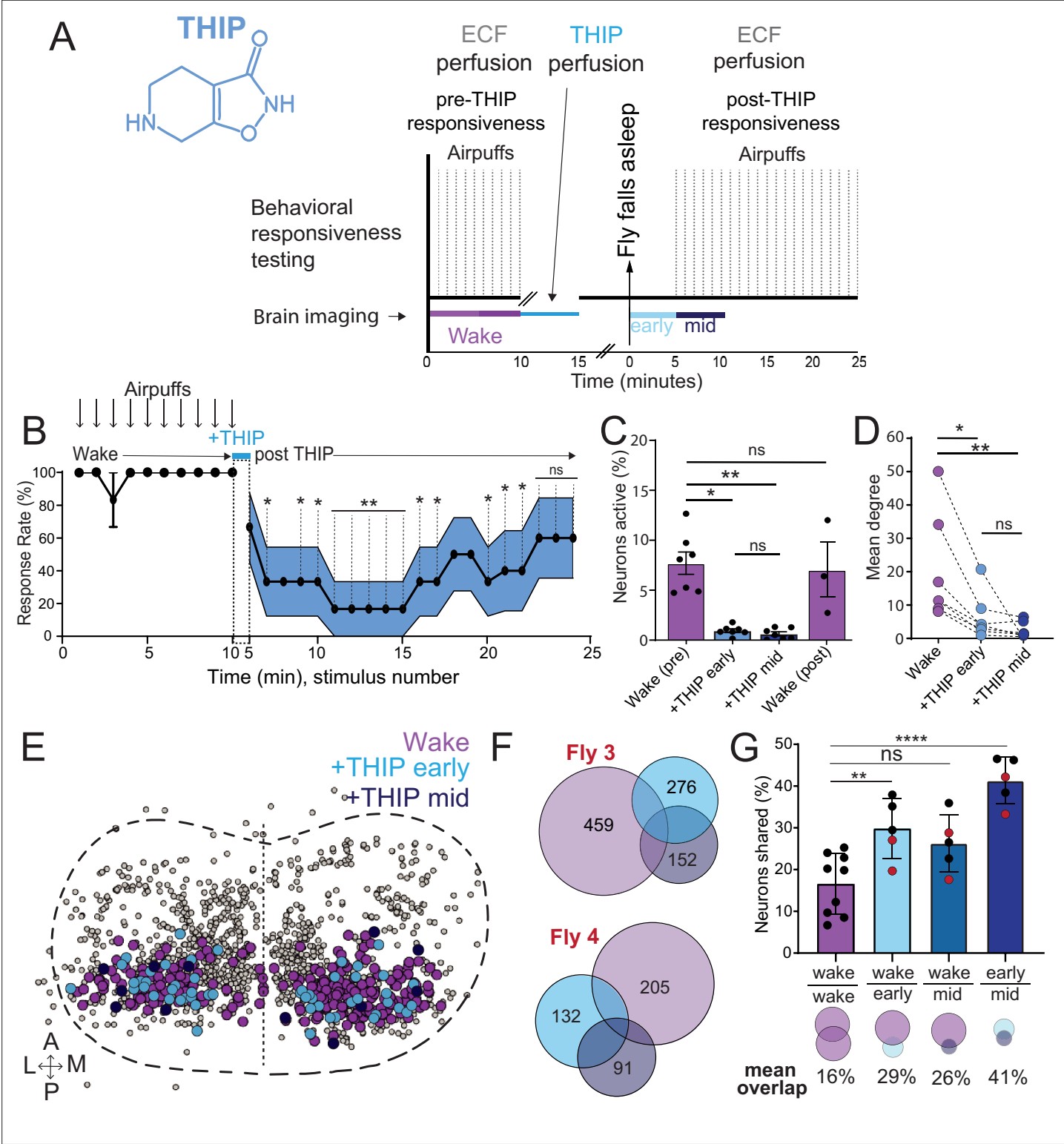

**Figure 5.** Brain activity and connectivity decrease during 4,5,6,7-tetrahyrdoisoxazolopyridin-3-ol (THIP)-induced sleep. (**A**) Experimental protocol for behavioral responsiveness and brain imaging experiments. 5 min of baseline condition were recorded, during which the exposed brain was perfused with extracellular fluid (ECF), followed by 5 min of THIP perfusion. Following sleep induction, an additional 10 min of calcium activity was recorded, which was separated into 'early' and 'mid' sleep for analysis. Air puff stimuli were delivered to test for behavioral responsiveness. (**B**) Mean behavioral response rate (% ± SEM) to air puff stimuli over the course of an experiment (n = 6). Air puff delivery times are indicated by the solid dots. (**C**) Percent neurons active (± SEM) in UAS:Chrimson/X; Nsyb:LexA/+; LexOp:nlsGCaMP6f/R23E10:Gal4 flies during wake, THIP-induced sleep, and recovery (n =

*Figure 5 continued on next page*

*Figure 5 continued*

9; three flies were recorded post-waking). (**D**) Correlation analysis (mean degree ± SEM) of active neurons in (**C**). (**E**) Collapsed mask of neurons active during wake, and both early and mid THIP sleep. (**F**) Overlap in neural identities between wake and THIP-induced sleep in two example flies. Number indicates active neurons within each condition; same color code as in (**E**). (**G**) Quantification of neural overlap data. Red dots indicate the flies shown in (**F**). n = 9 flies. All tests are one-way ANOVA with Dunnett's multiple-comparison test. ns, not significant, *p<0.05, **p<0.01, ****p< 0.001.

*2015*). We present our THIP results first. Since THIP is a GABA-acting drug that probably affects a variety of processes in the brain aside from sleep, we also assessed the effect of THIP on flies that were prevented from sleeping (*Figure 6A*, left panel). Sleep deprivation (SD) was performed by mechanically arousing flies once every 20 s for the duration of the experiment on a 'SNAP' apparatus (*Dissel et al., 2015*; *Seugnet et al., 2009*). RNA was extracted from the brains of all groups of flies (+/-THIP, +/-SD) after 10 hr of daytime (8 AM to 6 PM) THIP (or vehicle) provision. Samples for RNA-sequencing were collected in replicates of 5 to ensure accuracy, and any significant transcriptional effects were thresholded at a log fold change of 0.58 (see 'Materials and methods').

Flies allowed to eat food containing 0.1 mg/ml THIP ad lib over 10 daytime hours led to 129 significant changes in gene expression compared to vehicle-fed controls, with the large majority (110) being downregulated and only 19 upregulated (*Figure 6B, C and E*, *Figure 6—source data 1*). In contrast, when THIP-fed flies were prevented from sleeping, this led to mostly upregulated genes (88 upregulated vs. 21 downregulated, *Figure 6B, D and F*, *Figure 6—source data 2*), with many of the upregulated genes showing a high level of fold change in expression. Not surprisingly, preventing sleep in THIP-fed flies led to an almost entirely non-overlapping set of gene expression changes (*Figure 6B*). This suggests that the large number of downregulated genes in +THIP/SD flies pertain to sleep processes rather than the effect of ingesting THIP since only a few (9) genes overlapped with the +THIP/ + SD dataset, which revealed mostly upregulated genes.

Gene Ontology analysis on genes that were downregulated as a result of THIP-induced sleep highlighted a significant enrichment of metabolism pathways (*Figure 6E and F*, *Figure 6—figure supplement 1*). The top Gene Ontology biological processes included primary, organic substance, cellular, biosynthetic, and nitrogen compound metabolic pathways, as well as ribosomal processes. Interestingly, these downregulated processes are largely consistent with a recently published mouse sleep transcriptome study (*Muheim et al., 2019*). Among the metabolism pathways uncovered in this dataset, we observed over-representation of expected genes such as *bgm* (bubblegum CG4501) and *Acer* (angiotensin-converting enzyme-related CG10593). Both of these genes are found in the primary metabolic and organic substance metabolic processes as well as within the Sleep Gene Ontology dataset (GO:0030431). Another downregulated metabolic gene is AkhR (adipokinetic hormone receptor), which has been found to be involved in starvation-induced sleep loss in *Drosophila* (*He et al., 2020*). AkhR belongs to the Class A GPCR Neuropeptide and protein hormone receptors, which are a gene class involved in storage fat mobilization, analogous to the glucagon receptor found in mammals (*Bharucha et al., 2008*).

Although THIP-induced sleep overwhelmingly led to gene downregulation, a few genes (19) were significantly upregulated. Gene Ontology analysis on these upregulated genes highlighted enrichment in varying groups including developmental processes and multicellular organismal processes (*Figure 6E and F*, *Figure 6—figure supplement 1*). Some groups were enriched under the organic substance metabolic process pathways; however, there was no overlap when comparing these to the pathways enriched due to downregulation of genes. There were some overlapping enriched pathways when we compared the gene sets from sleep-deprived flies, which had also been treated with THIP (*Figure 6F*, *Figure 6—figure supplement 2*). However, the gene sets they involve are upregulated in the SD dataset but downregulated in sleeping flies. Interestingly, the non-sleeping THIP dataset uncovered a significant enrichment of pathways involved in the response to stress. This might be expected for flies exposed to regular mechanical stimuli over 10 hr. None of these pathways featured in the THIP sleep dataset.

To validate these findings, we conducted qRT-PCR analyses on 19 genes from our THIP sleep dataset and compared these results to our original transcriptional data. The genes represented a range of both up- and downregulated genes, and we found convincing correspondence between the groups (*Figure 5G*). qRT-PCR comparisons with RNA-seq data and associated statistics are presented in *Supplementary file 2*.

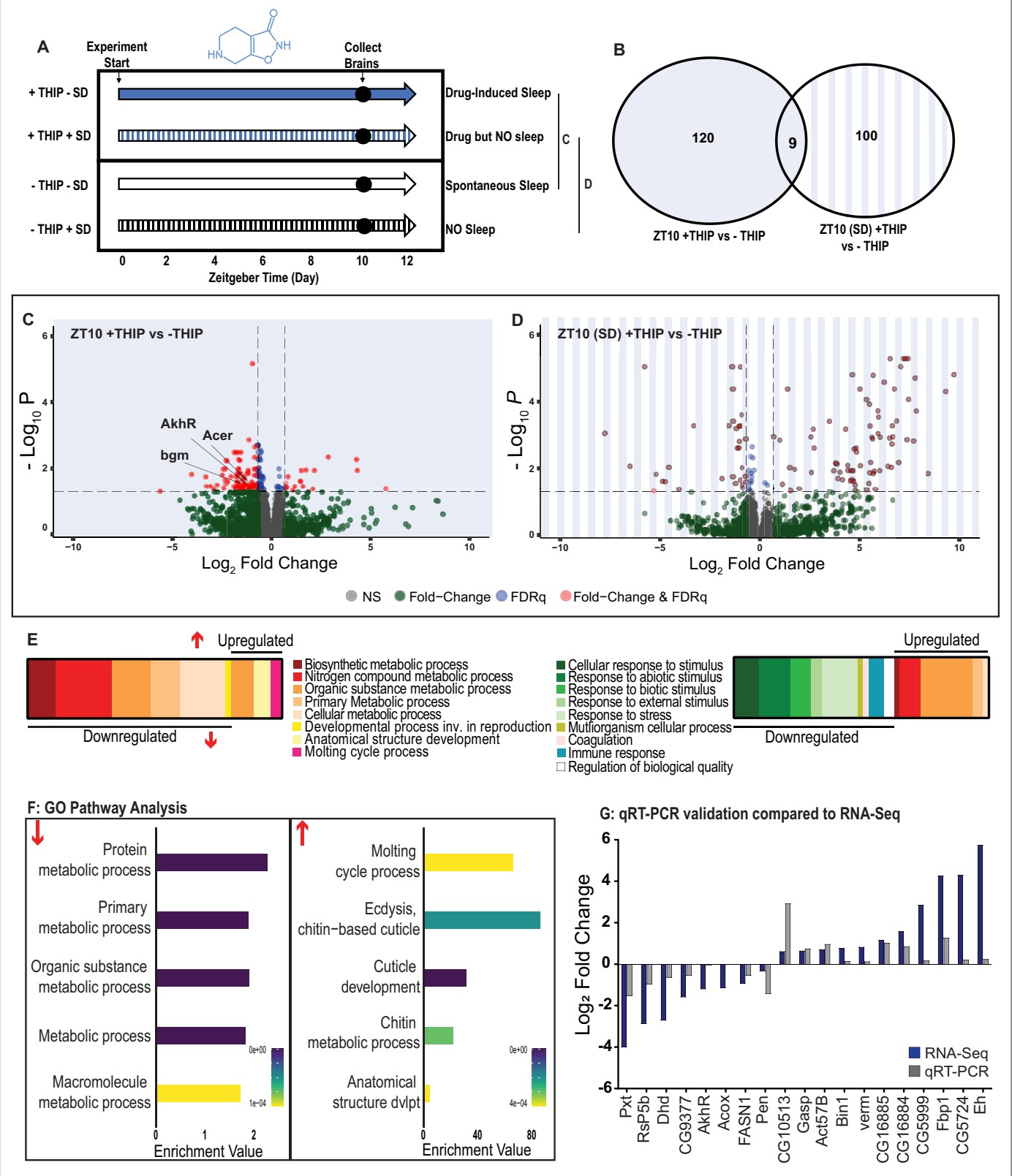

**Figure 6.** Metabolic processes are downregulated during 4,5,6,7-tetrahyrdoisoxazolopyridin-3-ol (THIP)-induced sleep. (**A**) Schematic representation of the experimental set-up and samples processed using RNA-sequencing. (**B**) Venn diagram showing the gene expression overlap between flies that had been treated with THIP vs. their control (shaded blue) and flies that had been treated with THIP in a sleep-deprived background vs. their control (shaded blue bars). The number of significant differentially expressed genes in each category is indicated. (**C**) Volcano plot representing the distribution

*Figure 6 continued on next page*

*Figure 6 continued*

of differentially expressed genes in the presence or absence of THIP. Genes that are significantly up-/downregulated meeting a Log2Fold change of 0.58 and FDRq value of 0.05 are shown in red. Genes meeting the threshold for FDRq value only are shown in blue. Fold change only is shown in green. Those genes not meeting any predetermined criteria are shown in gray. (**D**) Volcano plot representing the distribution of differentially expressed genes in the presence or absence of THIP in a sleep-deprived background. Criteria as above (**C**). (**E**) Schematic representation of Gene Ontology (GO) enrichment of biological process results. Color coded to indicate parent and child terms for comparisons between groups highlighted above (**C**, left, **D**, right). (**F**) Bar chart representation of a subset of interesting significant GO pathway terms originating from the organic substance and primary metabolic processes for the dataset shown in (**C**). (**G**) Comparison between significant gene hits obtained via RNA-sequencing (blue) and qRT-PCR (gray) in response to THIP, represented by Log2Fold change values. See *Figure 6—figure supplements 1 and 2* and *Figure 6—source data 1 and 2*.

The online version of this article includes the following source data and figure supplement(s) for figure 6:

**Source data 1.** List of significant 4,5,6,7-tetrahyrdoisoxazolopyridin-3-ol (THIP)-sleep genes.

**Source data 2.** List of significant sleep-deprivation genes in 4,5,6,7-tetrahyrdoisoxazolopyridin-3-ol (THIP)-fed flies.

**Figure supplement 1.** Gene Ontology (GO) enrichment analysis for 4,5,6,7-tetrahyrdoisoxazolopyridin-3-ol (THIP)-induced sleep.

**Figure supplement 2.** Gene Ontology (GO) enrichment analysis for 4,5,6,7-tetrahyrdoisoxazolopyridin-3-ol (THIP)-provisioned flies that were sleep deprived.

## Transcriptional analysis of flies induced to sleep by optogenetic activation

We next examined the effect of optogenetically induced sleep on the whole-brain transcriptome to compare to our THIP-induced sleep data. Based on our earlier findings that showed that optogenetic activation results in rapidly inducible sleep behavior that consolidates over at least 12 daytime sleep hours (*Figure 2C, E and G*), as well as our previous study showing that 10 daytime hours of optogenetic activation corrects attention defects in sleep-deprived flies (*Tainton-Heap et al., 2021*), we induced sleep in R23E10-Gal4 × UAS-Chrimson flies for 10 daytime hours and collected tissue for whole-brain RNA-sequencing (*Figure 7A*). We selected two timepoints for collection, for both the sleep-induced flies (+ATR) and their genetically identical controls that were not fed ATR (-ATR; *Figure 7A*). Optogenetic activation was matched to the normal daytime light cycle (8 AM to 8 PM). The first collection point was after 1 hr (ZT1, 9 AM) of red-light exposure to control for effects of ATR provision (when compared to ZT1, -ATR controls), as well as to uncover any potential short-term genetic effects of optogenetic activation. We then collected flies after 10 hr of red-light exposure (ZT 10, 6 PM) to examine longer-term genetic effects of optogenetic sleep induction and to match exactly our THIP sleep collection timepoint (i.e., 10 hr of induced daytime sleep by either method). The combined collection points also allowed us to compare transcriptional profiles between conditions (e.g., ZT10 +ATR vs. ZT10 -ATR), to identify sleep genes, as well as within conditions (ZT1 vs. ZT10), to account for genetic effects potentially linked to light-dark cycles. As for the THIP sleep data in the same strain, samples for RNA-sequencing were collected in replicates of 5 to ensure accuracy, and any significant transcriptional effects were thresholded at a log fold change of 0.58 (see 'Materials and methods').

We first examined the effect of 10 hr of daytime optogenetic-induced sleep. Here, we compared ATR-fed R23E10-Gal4 × UAS-Chrimson flies to genetically identical animals that were also exposed to red light for 10 hr but not provided with ATR in their food (ATR-). The control flies were therefore never induced to sleep by optogenetic activation, although they were still able to sleep spontaneously (see *Figure 2C, E and G*). We found that 10 hr of optogenetic activation led to 278 significant transcriptional changes, comprising mostly of upregulated genes, with 171 upregulated compared to 107 downregulated (*Figure 7B, C and E*, *Figure 7—source data 1*). In contrast to the THIP-induced sleep dataset, transcriptional analysis of 10 hr optogenetic sleep induction uncovered a variety of different processes predominantly related to the regulation of biological and cellular processes, rather than metabolism specifically (*Figure 7E*, *Figure 7—figure supplement 1*). For example, of the genes that were overexpressed there is an enrichment of the Semaphorin-plexin signaling pathway (GO:0071526, GO:1902287, GO:1902285, and GO:2000305) and the ephrin receptor signaling pathway (GO:0046011), both of which are known to be involved in axonal guidance (*Figure 7F*). Interestingly, several upregulated genes code for different subunits of nicotinic acetylcholine receptors (nAchRα1,3,4, and 5; *Figure 7C*). Importantly, there was almost no overlap with our SD dataset (*Figure 6—figure supplement 2, Figure 6—source data 2*), suggesting that optogenetic activation is

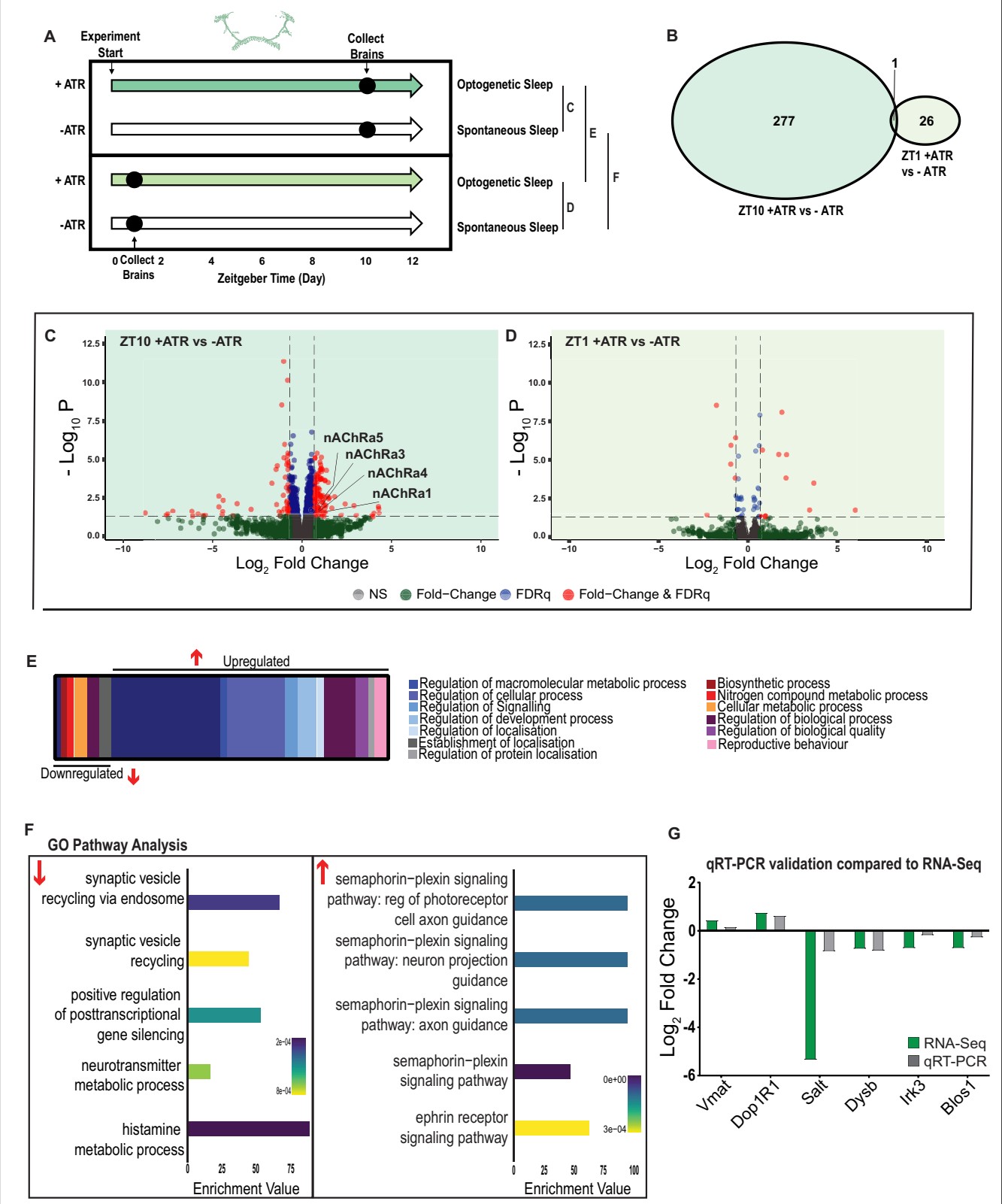

**Figure 7.** A variety of biological processes including axon guidance are upregulated during optogenetic-induced sleep. (**A**) Schematic representation of the experimental set-up and samples processed using RNA-sequencing. (**B**) Venn diagram showing the gene expression overlap between flies that experienced 10 hr of optogenetic-induced sleep (ZT10) compared to -ATR controls (ZT10) and those flies where optogenetic activation was restricted to 1 hr (ZT1) and compared to -ATR controls (ZT1). (**C**) Volcano plot representing the distribution of differentially expressed genes resulting

*Figure 7 continued on next page*

*Figure 7 continued*

from optogenetic optogenetic activation for 10 hr vs. control flies that were allowed to sleep spontaneously for 10 hr. Genes that are significantly up-/downregulated meeting a Log2Fold change of 0.58 and FDRq value of 0.05 are shown in red. Genes meeting the threshold for FDRq value only are shown in blue. Fold change only in green. Those genes not meeting any predetermined criteria are shown in gray. (**D**) Volcano plot representing the distribution of differentially expressed genes resulting from optogenetic activation for 1 hr vs. control flies that were allowed to sleep spontaneously for 1 hr. Criteria as above (**C**). (**E**) Schematic representation of Gene Ontology (GO) enrichment of biological process results. Color coded to indicate parent and child terms comparing flies that had been activated optogenetically for 10 hr vs. flies that had been allowed to spontaneously sleep for the same duration. (**F**) Bar chart representation of a subset of interesting significant GO pathway terms originating from the regulation of cellular processes and signaling biological processes. (**G**) Comparison between significant gene hits obtained via RNA-sequencing (green) and qRT-PCR (gray) in response to optogenetic sleep, represented by Log2Fold change values. See *Figure 7—figure supplements 1–3* and *Figure 7—source data 1–4*.

The online version of this article includes the following source data and figure supplement(s) for figure 7:

**Source data 1.** List of significant optogenetic-sleep genes after 10 hr activation.

**Source data 2.** List of significant optogenetic-sleep genes after 1 hr activation.

**Source data 3.** List of significant ZT10 vs. ZT1 genes in ATR+ dataset.

**Source data 4.** List of significant ZT10 vs. ZT1 genes in ATR- dataset.

**Figure supplement 1.** Gene Ontology (GO) enrichment analysis for optogenetic-induced sleep.

**Figure supplement 2.** Circadian-related genes uncovered in optogenetic-sleep dataset.

**Figure supplement 3.** Summary of different Gene Ontology (GO) pathways engaged by optogenetic-induced sleep and 4,5,6,7-tetrahyrdoisoxazolopyridin-3-ol (THIP)-induced sleep.

not sleep depriving (or stressing) the flies (only one upregulated gene was shared, CG40198). Of the genes that were downregulated, there is enrichment of pathways that relate to synaptic vesicle recycling (GO:0036465 and GO:0036466) as well as neurotransmitter metabolic processes (GO:0042133) (*Figure 7F*, *Figure 7—figure supplement 1*).

In contrast to the 10 hr timepoint, 1 hr of optogenetically induced sleep had far fewer transcriptomic consequences, with only 17 genes upregulated and 10 downregulated (*Figure 7B and D*). This small number of transcriptomic changes (see *Figure 7—source data 2*) may reflect the effect of ATR feeding, rather than any genes relevant to optogenetic sleep. That 9 hr of additional optogenetic sleep increased transcriptomic changes by an order of magnitude lends confidence to the interpretation that relevant genes linked to prolonged optogenetic activation are being engaged.

To account for potential genetic effects linked to light-dark expression cycles, we compared transcriptional profiles between 10 hr of optogenetic-induced sleep to 1 hr of induced sleep. Here, we found 220 differentially regulated genes (119 upregulated and 101 downregulated) when comparing ATR-fed flies at both timepoints (ZT10 vs. ZT1, *Figure 7—source data 3*). Since the 1 hr group was collected in the morning and the 10 hr group was collected in the evening, we expected this dataset to expose a number of circadian-regulatory genes, and this is indeed what we found (*Figure 7—figure supplement 2*). We then compared these results with a parallel ZT10 vs. ZT1 experiment where flies were not fed ATR. Here we uncovered 503 differentially expressed genes (252 upregulated and 251 downregulated) when comparing flies that had not been fed ATR at both timepoints (*Figure 7—source data 4*). Importantly, there were 98 genes that overlapped between these independent Z10 vs. ZT1 datasets, suggesting commonalities linked to circadian processes. Indeed, GO Pathway analysis of Biological Processes revealed a number of genes involved in the regulation of the circadian rhythm among these 98 overlapping genes, including the well-known circadian genes *period, timeless, clockwork-orange, clock and vrille*. Notably, co-factors *period* and *timeless* are both upregulated whereas *clk* is downregulated, and this is replicated in both independent datasets (*Figure 7—figure supplement 2*). This correspondence with expectations for zeitgeber or light-dark effects provides a level of confidence that our respective sleep datasets are highlighting transcriptomic changes and biological pathways relevant to either sleep induction approach. Notably, there was no overlap at all in gene expression changes between optogenetic-induced sleep and THIP-induced sleep (*Figure 6—figure supplement 1* vs. *Figure 7—figure supplement 1*; *Figure 6—source data 1* vs. *Figure 7—source data 1*), and the respective GO pathways analyses of biological processes are also largely non-overlapping (*Figure 7—figure supplement 3*).

To validate these findings, we compared our transcriptional results with qRT-PCR on six genes. This included the dopamine receptor Dop1R1, which regulates arousal levels (*Ferguson et al., 2017*)

as well as the schizophrenia-susceptibility gene dysbindin (*Dysb*), which has been shown to regulate dopaminergic function (*Shao et al., 2011*). We found convincing correspondence between our qRT-PCR data and our transcriptomic data (*Figure 7G*). qRT-PCR comparisons with RNA-seq data and associated statistics are presented in *Supplementary file 2*.

## Nicotinic acetylcholine receptors regulate sleep architecture

While THIP-induced sleep caused a systemic downregulation of metabolism-related genes, the effect of optogenetic-induced sleep on gene expression was not dominated by a single category. This may be consistent with our earlier observation that brain activity looks similar to wake during optogenetic-induced sleep (*Tainton-Heap et al., 2021*), so we could essentially be highlighting biological processes relevant to an awake fly brain, such as dopamine function (*Van Swinderen and Andretic, 2011*). However, optogenetic activation of the R23E10 circuit is not like wake, in that flies are rendered highly unresponsive to external stimuli, so perhaps like REM sleep in mammals a different category of molecular processes could be involved. In mammals, acetylcholine generally promotes wakefulness and alertness, but activity of cholinergic neurons is also high during REM sleep (*Watson et al., 2010*). Neurotransmission in the insect brain is largely cholinergic, with seven different nicotinic 'alpha' receptor subunits expressed in neural tissue (*Rosenthal and Yuan, 2021*). Interestingly, four of these subunits were significantly upregulated in our optogenetic sleep dataset: nAchRα1, nAchRα3, nAchRα4, and nAchRα5 (with nAchRα6 and nAchRα7 approaching significance, *Figure 7—source data 1*). For comparison, none of these were upregulated in our SD dataset, suggesting a sleep-relevant role. Previous studies have demonstrated a role for some of these same receptor subunits in sleep regulation, in particular nAchRα4 (also called *redeye*), which is upregulated in short-sleeping mutants (*Shi et al., 2014*), and nAchRα3, which has been reported to regulates arousal levels in flies (*Dai et al., 2021*). Together, these studies suggest processes that might be reconsidered in the context of active sleep, as highlighted by our gene expression findings. We therefore sought to examine the role of cholinergic signaling in sleep more closely by knocking out each nAchRα subunit and examining the effects of each subunit knockout on sleep architecture. Since our transcriptomic analysis encompassed effects of active sleep on whole-brain gene expression, we first eliminated each nAchRα subunit across the brain by testing confirmed genetic deletions (*Chen et al., 2022*; *Perry et al., 2021*).

We first examined the effect of each nAchRα subunit deletion on sleep duration using the 5 min criterion for quantifying sleep in *Drosophila* (*Shaw et al., 2000*). We found that the nAchRα mutants fell into two different categories: removal of nAchRα1 and nAchRα2 significantly decreased sleep, day and night; whereas removal of nAchRα3, nAchRα4, nAchRα6, and nAchRα7 significantly increased sleep, day and night (*Figure 8A*). The nAchRα5 knockout was found to be homozygous lethal, so was not included. To examine sleep architecture in these mutants, we quantified sleep bout number and duration and plotted these together as done previously for our sleep induction experiments (*Figure 2*). Examining the data this way, it is clear to see how nAchRα1 and nAchRα2 are different: most sleep bouts are very short, day and night (*Figure 8B*, top two rows, left panels, green dots). In contrast, knocking out the other alpha subunits seems to consolidate sleep, especially at night (*Figure 8B*, bottom four rows, left panels). nAchRα3 knockouts were most striking in this regard, with these flies sleeping uninterrupted for an average of 156.53 min (± 18.06) during the day and 160.76 min (± 17.92) at night. Increased sleep consolidation in these mutants was, however, not due to the lack of activity. While awake, nAchRα3 animals were just as active as controls (activity per waking minute = 2.69 ± 0.18 vs. 2.5 ± 0.06, respectively). However, some knockouts did increase waking activity levels; waking activity data for all of the knockout strains and their genetic controls are presented in *Figure 8—figure supplement 1*.

We next questioned what kind of sleep the nAchRα knockout flies might be getting. In previous work, we have shown that flies can be asleep already after the first minute of spontaneous inactivity, and that during the first 5 min of sleep the fly brain displays wake-like levels of neural activity (*Tainton-Heap et al., 2021*). We have termed this early sleep stage 'active sleep' to distinguish it from 'quiet sleep' that typically follows after 5–10 min (*van Alphen et al., 2013*). One way of estimating the amount of 'active sleep' in *Drosophila* flies is to sum all short sleep epochs lasting between 1 and 5 min and expressing this as a percentage of total sleep (*Tainton-Heap et al., 2021*). When we re-examined our nAchRα knockouts in this way, we found that this behavioral readout for 'active sleep' was

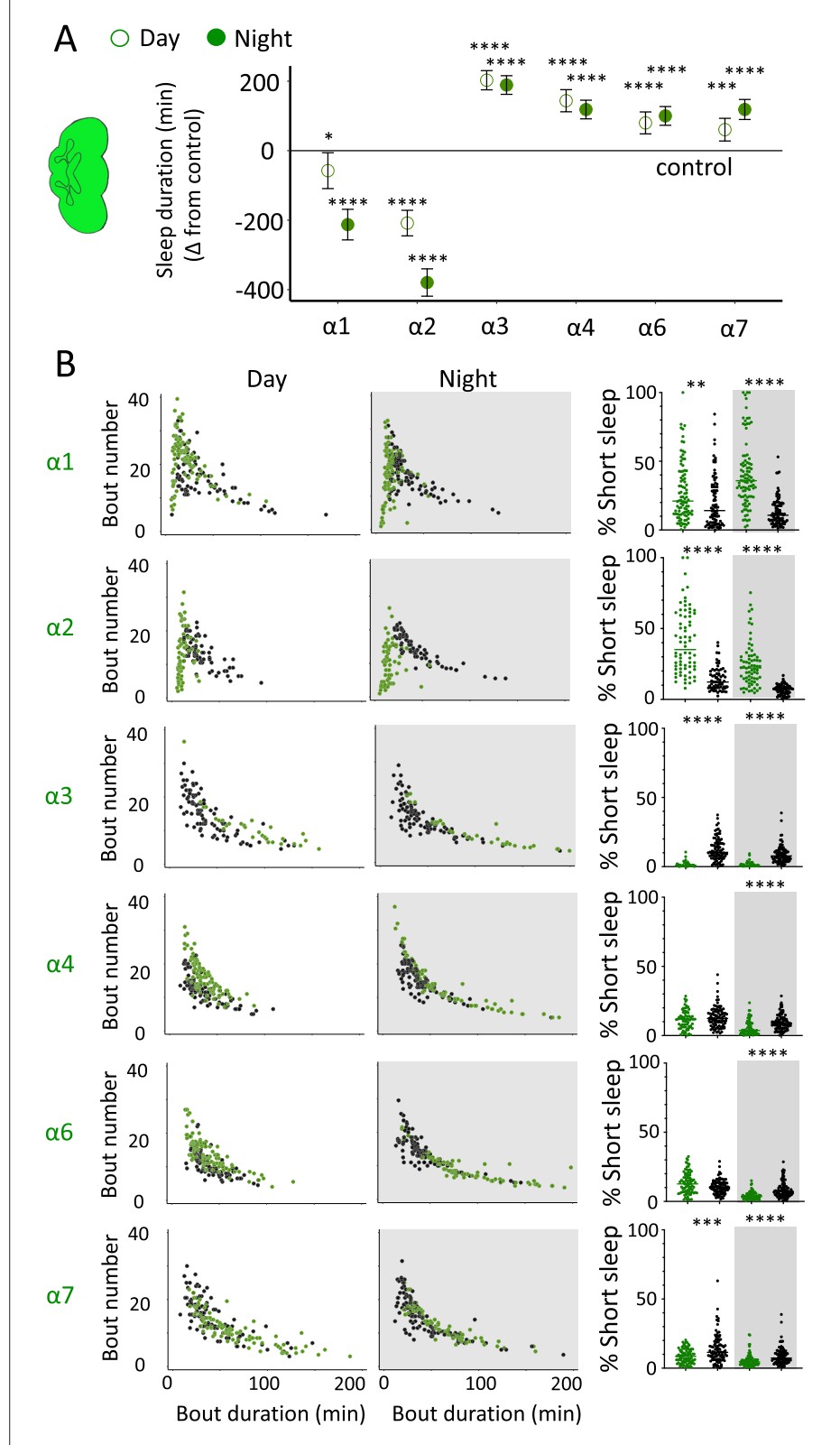

**Figure 8.** nAchRα subunit knockouts bidirectionally regulate >5 min sleep as well as short sleep. (**A**) Average total day and night sleep duration (minutes ± 95% confidence intervals) in nAchRα knockout mutants, expressed as difference to their respective background controls (see 'Materials and methods'). α1, N = 91; contro1 ($X^{59}w^{1118}$) = 93; α2, N = 70; control ($w^{1118}$ActinCas9) = 65; α3, N = 43; (ActinCas9) = 91; α4, N = 87; ($w^{1118}$ActinCas9) = 98; α6,

*Figure 8 continued on next page*

*Figure 8 continued*

N = 91; (w$^{1118}$ActinCas9) = 91; α7, N = 94; (ActinCas9) = 95. *p<0.05, ***p<0.001, ****p<0.0001 by *t*-test adjusted for multiple comparisons. (**B**) Left two panels: sleep architecture for the same six knockout strains as in (**A**) (green), shown against their respective controls (black). Each datapoint is a fly. Right panels: cumulative short sleep (1–5 min) expressed as a percentage of total sleep duration. Data are from the same experiment as in (**A, B**). Each datapoint is a fly. **p<0.01, ***p<0.001, ****p<0.0001 Mann–Whitney *U* test. All data were collected over 3 days and 3 nights and averaged.

The online version of this article includes the following figure supplement(s) for figure 8:

**Figure supplement 1.** Waking activity levels of nAChRα knockout mutants.

significantly affected by the loss of select nAchRα subunits. Short sleep increased significantly during both the day and the night in nAchRα1 and nAchRα2 (*Figure 8B*, top two rows, right panel, green dots). In contrast, and consistent with our sleep architecture analyses (above), nAchRα3 displayed almost no short sleep (*Figure 8B*, row 3, right panel). Finally, in nAchRα4 and nAchRα6 short sleep was significantly decreased at night, while in nAchRα7 short sleep was significantly decreased day and night (*Figure 8B*, rows 4–6, right panel). In conclusion, every one of the nAchRα knockouts we tested affect short sleep in some way, either increasing (nAchRα1 and nAchRα2) or decreasing it (nAchRα3, nAchRα4, nAchRα6, nAchRα7). Additionally, tallying the proportion of short sleep bouts provides valuable insight into altered sleep architecture in mutant strains.

We nevertheless questioned whether these systemic effects of nicotinic receptors on short sleep were perhaps a trivial consequence of altered >5 min sleep duration in these mutants, especially regarding the striking differences between nAchRα1&2 and the other subunit knockouts. We therefore returned to our 'quiet' sleep (THIP) dataset to contrast a gene derived from that study. We found that several of the THIP-induced sleep genes are involved in metabolic processes, which are mostly downregulated (*Figure 6—figure supplement 1*, *Figure 6—source data 1*). This included the adipokinetic hormone receptor (AkhR), which has previously been associated with starvation-induced sleep regulation (*He et al., 2020*). We employed an RNAi strategy to downregulate this metabolic gene's expression across the fly brain in AkhR-RNAi/R57C10-Gal4 flies (see 'Materials and methods'). We found that downregulating AkhR significantly decreased sleep duration during the day as well as night compared to genetic control strains (*Figure 9A and B*). Accordingly, sleep bout duration and number decreased, especially during the day (*Figure 9C*). However, in contrast to knocking out nAchRα1 and nAchRα2, which also significantly decreased sleep duration day and night, short sleep was not significantly altered in AkhR knockdown animals compared to genetic controls (*Figure 9D*). This suggests that short (1–5 min) sleep might be under separate regulatory control than >5 min sleep.

We then returned to the nAchRα subunits and employed an RNAi strategy to knock them down in the R23E10 sleep-promoting neurons specifically. While it is unlikely that these cholinergic receptor subunits are restricted to just these neurons, we were curious if removing any of them from this specific circuit had similar effect on sleep as in the knockout strains. We found that nAchRα1 knockdown in the R23E10 neurons significantly decreased total sleep during both day and night (*Figure 10A and B*), which was also true for the nAchRα1 knockouts (*Figure 8A*). Also consistent with the knockouts, the proportion of short sleep bouts was increased in this specific knockdown (*Figure 10C*). Among all of the other alpha subunits, knockdown of nAchRα5 and nAchRα6 in the R23E10 neurons significantly increased daytime and night-time sleep, respectively (*Figure 10A and B*), while short sleep was correspondingly decreased only in nAchRα5 (*Figure 10C*). None of the other alpha subunits had any effect on these sleep-promoting neurons. This suggests that their sleep-regulatory role lies outside of this circuit, while nAchRα1, nAchRα5, and nAchRα6 might regulate sleep-related functions within this circuit.

## Discussion

One of the key advantages of studying sleep in *Drosophila* is that this versatile model provides a variety of reliable approaches for inducing and controlling sleep. By being able to induce sleep on demand, either genetically or pharmacologically, researchers have been able to manipulate sleep as an experimental variable, and in this way be better able to assess causality when probing potential sleep functions. However, this approach has often sidestepped the question of whether different sleep

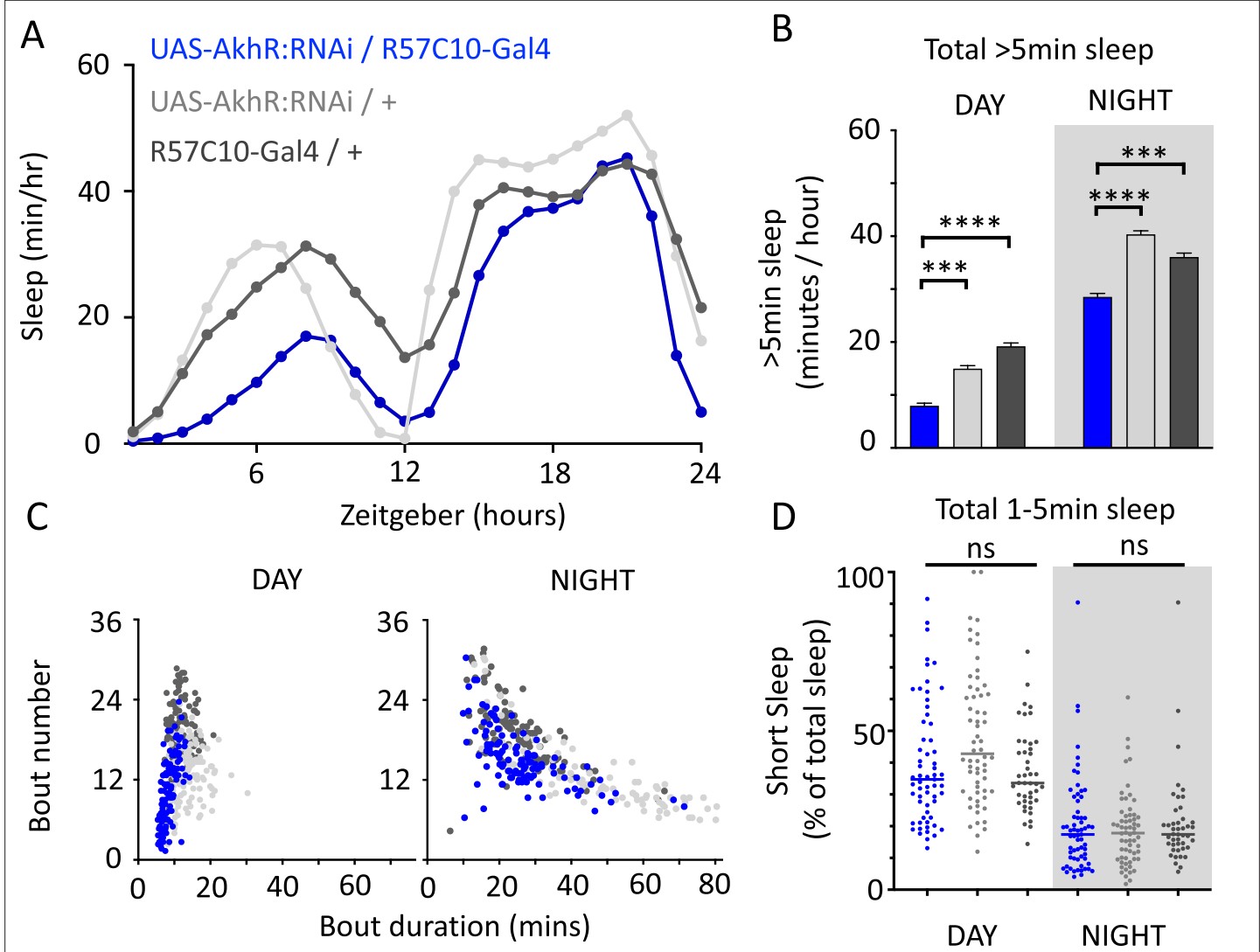

**Figure 9.** AkhR knockdown decreases >5 min sleep but not short sleep. (**A, B**) Total sleep (>5 min) in UAS-AkhR:RNAi/R57C10-Gal4 flies (blue, N = 126) compared to genetic controls (light gray: UAS-AkhR:RNAi/+, N = 124; dark gray: R57C10-Gal4/+, N = 120). (**C**) Sleep architecture (average bout duration vs. bout number per fly) in data from (**A, B**). (**D**) Cumulative short sleep (1–5 min, expressed as a % of total sleep) in UAS-AkhR:RNAi/R57C10-Gal4 flies (blue) compared to genetic controls (light gray: UAS-AkhR:RNAi/+; dark gray: R57C10-Gal4/+). Wild-type background (+) is Canton-S(w[1118]). Each datapoint is a fly. ***p<0.001, ****p<0.0001 Mann–Whitney *U* test. ns, not significant. All data were collected over 2 days and 2 nights and averaged.

induction methods are equivalent or whether distinct forms of sleep might be engaged by different genetic or pharmacological treatments. In mammals, GABA agonists typically promote SWS, which has been associated with cellular homeostasis and repair process in the brain (*Tononi and Cirelli, 2014*; *Xie et al., 2013*). In contrast, drugs targeting acetylcholine receptors, such as carbachol, have been found to promote brain states more reminiscent of REM sleep (*Kubin, 2001*; *Torterolo et al., 2016*). Although these drugs all induce sedative (or dissociative) states, they are clearly producing dissimilar forms of sleep in mammals, with likely different functions or consequences for the brain. In *Drosophila*, evidence suggests that the GABA agonist THIP promotes a form of deep or 'quiet' sleep, which may be functionally analogous to mammalian SWS (*Yap et al., 2017*; *van Alphen et al., 2021*; *Kirszenblat and van Swinderen, 2015*). In contrast, optogenetic activation of the dFB as well as other neurons may promote a form of active sleep, which we have suggested could be achieving some similar sleep functions as REM (*Van De Poll and van Swinderen, 2021*). THIP-induced sleep in flies has been associated with waste clearance from the brain (*van Alphen et al., 2021*), just as SWS has been associated with clearance of waste metabolites via the mammalian brain's glymphatic system

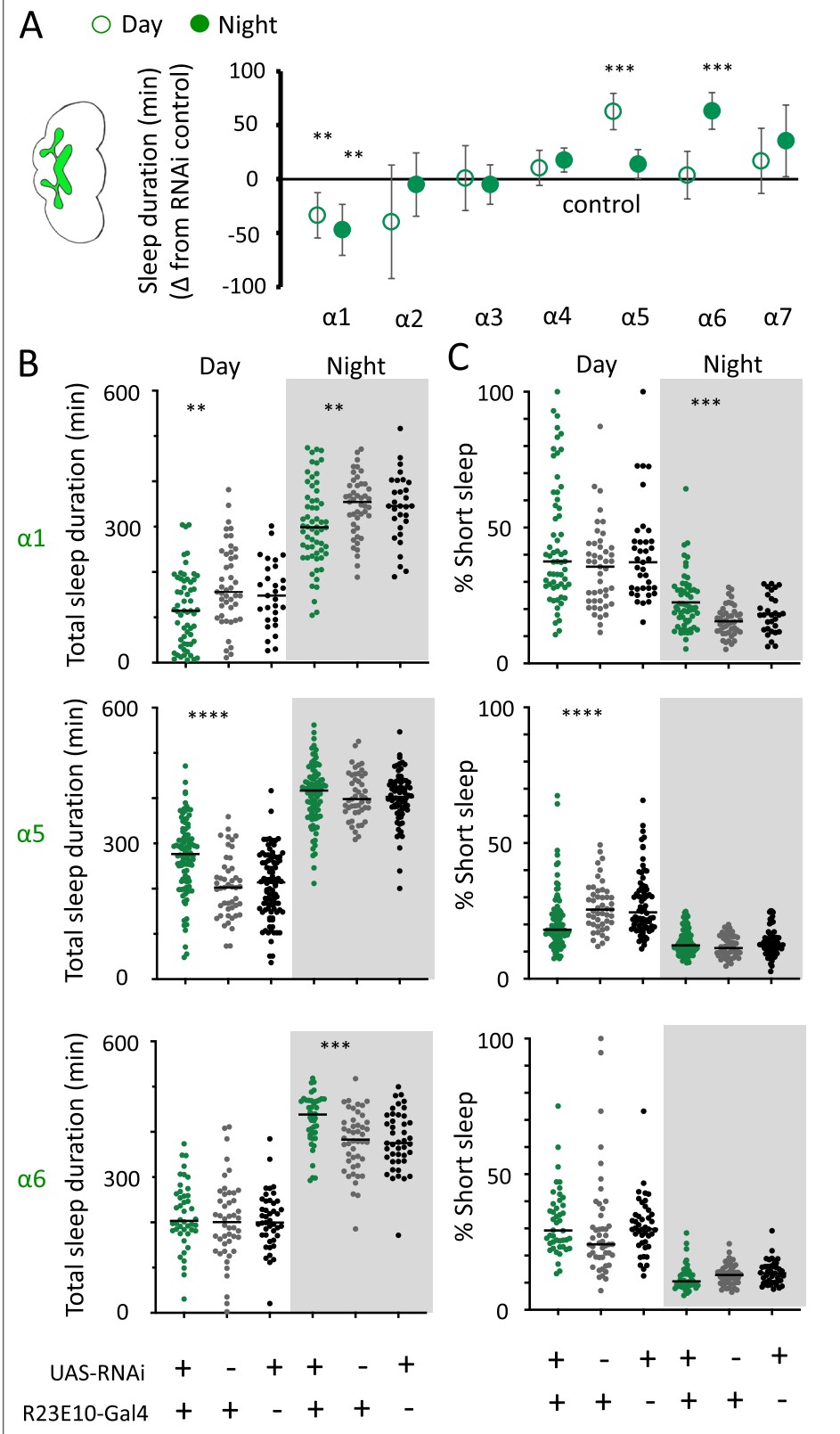

**Figure 10.** nAchRα subunit knockdowns in sleep-promoting neurons. (**A**) Average total day and night sleep duration (minutes ± 95% confidence intervals) in UAS-nAchRα RNAi/R23E10-Gal4 flies, expressed as difference to RNAi/+controls (all were also compared Gal4/+controls, not shown). α1 N = 60, Gal4 N = 45, RNAi N = 37; α2 N = 21, Gal4 N = 24, RNAi N = 50; α3 N = 33, Gal4 N = 50, RNAi N = 59; α4 N = 92, Gal4 N = 90, RNAi N =

*Figure 10 continued on next page*

*Figure 10 continued*

80; α5 N = 94, Gal4 N = 47, RNAi N = 74; α6 N = 44, Gal4 N = 47, RNAi N = 43; α7 N = 32, Gal4 N = 57, RNAi N = 44. **p<0.01, ***p<0.001 by *t*-test adjusted for multiple comparisons. (**B**) Total >5 min sleep duration data for significant knockdowns in (**A**). (**C**) Cumulative short sleep (1–5 min) expressed as a percentage of total sleep duration. Data are from the same experiment as in (**A, B**). Each datapoint is a fly. ***p<0.001, ****p<0.0001 Mann–Whitney *U* test. All data were collected over 2 days and 2 nights and averaged.

(*Xie et al., 2013*). Such functional homology suggests that the transcriptional changes we uncovered for THIP-induced sleep in *Drosophila* might also be relevant for mammalian SWS, with these largely centered on reduced metabolic processes and stress regulation (*Muheim et al., 2019*). In contrast, except for the upregulation of cholinergic signaling (*Vazquez and Baghdoyan, 2001*), there is little to compare to test hypotheses potentially linking active sleep in flies with REM sleep in mammals, except for the potential upregulation of cholinergic signaling. Even this cholinergic connection seems odd, seeing that the predominant excitatory neurotransmitters are reversed in the brains of insects and mammals: glutamate in mammals and acetylcholine in insects (*Smarandache-Wellmann, 2016*). Additionally, only nicotinic receptor subunits were identified in our analyses, whereas muscarinic receptors have been more commonly associated with REM sleep in mammals (*Bourgin et al., 1995*; *Kashiwagi and Hayashi, 2018*). Nevertheless, it is clear from our results that optogenetic-induced active sleep upregulates the expression of multiple nicotinic acetylcholine subunits, and that knocking these out individually has profound (and opposing) effects on sleep architecture in flies, and that a subset might function intrinsically within sleep-promoting neurons. This supports other studies showing a role for nAchRα's in sleep regulation (*Shi et al., 2014*; *Dai et al., 2021*). It will be especially interesting in future brain imaging studies to see whether a knockout such as nAchRα3 is eliminating one kind of sleep (e.g., active sleep) as predicted by our behavioral data, and whether this is associated with any functional consequences. Similarly, it will be telling to see whether the opposite sleep phenotypes observed in nAchRα1, for example, result in a distinct class of functional consequences. A previous study has shown that nAchRα1 knockout animals have significantly shorter survivorship compared to controls, with flies dying almost 20 d earlier (*Somers et al., 2017*). One reason could be because of impaired or insufficient deep sleep functions (e.g., brain waste clearance; *van Alphen et al., 2021*). The nAchRα knockouts provide an opportunity to further examine mutant animals potentially lacking either kind of sleep, although this will have to be confirmed by brain imaging or electrophysiology.

We found little similarity between two different approaches to inducing sleep in flies at the level of gene expression as well as acute effects on brain activity. It may, however, not be surprising that these entirely different sleep induction methods produce dissimilar physiological effects. After all, one method requires flies to ingest a drug, which then must make its way to the brain, while the other method directly activates a subset of neurons in the central brain, as well as in the ventral nerve cord (*Jones et al., 2023*). Yet both methods yield similarly increased sleep duration profiles and consolidated sleep architecture (*Figure 2*). One underlying assumption with focusing on sleep duration as the most relevant metric for understanding sleep function in *Drosophila* is that sleep is a unitary phenomenon in the fly model, meaning that primarily one set of functions and one form of brain activity are occurring when flies sleep. There is now substantial evidence that this is unlikely to be true, and that like other animals flies probably also experience distinct sleep stages that accomplish different functions (*Yap et al., 2017*; *Tainton-Heap et al., 2021*; *van Alphen et al., 2021*; *van Alphen et al., 2013*; *Wiggin et al., 2020*; *Gong et al., 2022*). This does not mean that these functions are mutually exclusive; for example, both THIP provision and neuronal activation of central brain circuits have been found to promote memory consolidation in *Drosophila* (*Dissel et al., 2015*). Indeed, it seems reasonable to propose that different sleep stages could be synergistic, accomplishing a variety of homeostatic functions that might be required for adaptive behaviors in an animal. Our results suggest that THIP provision promotes a 'quiet sleep' stage in flies, which induces a brain-wide downregulation of metabolism-related genes. This is consistent with studies in flies showing that metabolic rate is decreased in longer sleep bouts, especially at night, and that this is recapitulated by THIP-induced sleep (*Stahl et al., 2017*). Our findings also align with a previous study showing that mitochondrial metabolic processes are upregulated in the absence of sleep (*Kempf et al., 2019*). One argument for why metabolism-related genes are downregulated during THIP-induced sleep might be that flies are starved (because they are sleeping more, and thus perhaps not eating). However, flies induced

to sleep by optogenetic activation are also sleeping more (indeed, their average sleep bouts are longer), yet these did not reveal a similar downregulation of metabolic processes. Another view might be that our sampling was done after flies had achieved 10 hr of induced sleep, so sleep functions might have already been achieved by that time. Thus, we might not be uncovering genes required for achieving 'quiet' sleep functions as much as identifying exactly the opposite: genetic pathways that have been satisfied by 10 hr of induced quiet sleep. Other studies using THIP to induce sleep have examined longer time frames (e.g., 2 d; *Dissel et al., 2015*), so it remains unclear whether changes in gene regulation relate to sleep functions that have been achieved or that are still being engaged. For our transcriptomic study, both sleep induction protocols were conducted during the day (thus in presumably well-rested flies) to deliberately avoid any sleep pressure confounds. Since the dFB has been proposed as a regulator of sleep homeostasis (*Donlea et al., 2011*; *Donlea et al., 2014*; *Donlea et al., 2018*), one could question why its prolonged activation did not incur a deep sleep debt, which might have been detectable at a transcriptomic level by some overlap with our SD controls. That we did not see this suggests induced daytime active sleep might not accrue any sleep debt. We have shown previously that daytime sleep is qualitatively different than night-time sleep in flies, which is 'deeper' (*van Alphen et al., 2013*). It is, for example, possible that most quiet sleep happens at night in flies, so a daytime regime of induced active or quiet sleep might not incur any sleep debt for either. Following this logic, our RNA analyses simply revealed the transcriptomic changes resulting from either manipulation in well-rested flies, with the key result being that there are zero shared genes after 10 daytime hours of either manipulation.

In contrast to THIP, our optogenetic manipulation induced a brain-wide upregulation of a variety of neural mechanisms unrelated to metabolic processes, suggesting a different kind of sleep was engaged. Although many studies have shown that the R23E10-Gal4 circuit is sleep promoting (e.g., *Troup et al., 2018*; *Donlea et al., 2014*; *Gong et al., 2022*), and we have shown that acute activation of these neurons produces a form of active sleep (*Tainton-Heap et al., 2021*), it seems unlikely that active sleep regulation is limited to the dFB neurons alone (*Jones et al., 2023*). Other circuits in the fly central brain are also sleep-promoting, including in the ellipsoid body (*Liu et al., 2016*) and the ventral fan-shaped body (vFB) (*Lei et al., 2022*), although it remains unknown whether activation of these other circuits also promotes an active sleep stage or whether a similar transcriptome might be engaged by these alternate approaches to optogenetic sleep induction in flies. This again highlights a variant of the same problem we have uncovered in this study comparing pharmacology with optogenetics: different circuit-based approaches could all be increasing sleep duration but achieving entirely different functions by engaging distinct transcriptomes and thus different sleep functions. How many different kinds of functions are engaged by sleep remains unclear: is it roughly two functional categories linked to quiet and active sleep, or is it a broader range of subcategories that are not so tightly linked to these obviously different brain activity states?

A compelling argument could nevertheless be made for two kinds of sleep in most animals, with two distinct sets of functions (*Jaggard et al., 2021*; *Kirszenblat and van Swinderen, 2015*). Most animals have been shown to require a form of 'quiet' sleep to ensure survival, suggesting that these might encompass an evolutionarily conserved set of cellular processes that promote neural health and development (*Zimmerman et al., 2008a*), and that operate best during periods of behavioral quiescence. Nematode worms thus experience a form of quiet sleep when they pause to molt ('lethargus') into different life stages during their development (*Raizen et al., 2008*) or when cellular repair processes are needed following environmental stress (*Hill et al., 2014*). In flies, quiet sleep seems to be similarly required for neuronal repair (*Stanhope et al., 2020*) or waste clearance (*van Alphen et al., 2021*), and there is evidence that glia might play a key role in these cellular homeostatic processes in flies (*Stanhope et al., 2020*), as well as other animals (*Artiushin and Sehgal, 2020*). Thus, SWS in mammals and birds might present a narrow neocortical view of a more ancient set of sleep functions centered on quiescence and decreased metabolic rate. Indeed, neural quiescence is also a feature of SWS, both at the level of pulsed inhibition (down-states) and in other parts of the brain beyond the cortex (*Jaggard et al., 2021*). Similar to findings in flies that are induced into a quiet sleep stage with THIP (*Stahl et al., 2017*), metabolic rate also decreases during SWS in mammals (*Caron and Stephenson, 2010*). In contrast, metabolic rate is similar to waking during REM sleep in mammals (*Maquet, 1995*), suggesting an alternate set of functions not linked to cellular homeostasis. What might these functions be, and could some of these be conserved between active sleep in invertebrates and REM sleep

in mammals? A REM-like sleep stage has now been identified in a variety of invertebrate species, including cephalopods (*Iglesias et al., 2019*) and jumping spiders (*Rößler et al., 2022*). In humans, REM sleep has been implicated in emotion regulation (*Hutchison and Rathore, 2015*), and cognitive disorders where emotions are dysregulated, such as depression, are often associated with REM sleep dysfunction (*Harrington et al., 2018*). While it is not evident how to study emotions in insects (but see *Anderson and Adolphs, 2014*), it could be argued that arousal systems more generally are employed to detect prediction errors and thereby promote learning (*Seth and Friston, 2016*). Thus, we and others have suggested that REM sleep might be crucial for optimizing prediction, and attention, and learning (*Van De Poll and van Swinderen, 2021*; *Kirszenblat and van Swinderen, 2015*; *Hobson, 2009*), and this may involve different kinds of homeostatic mechanisms centered on brain circuits rather than cells. Whether active sleep in flies plays a similar homeostatic role remains to be determined. Our finding that optogenetically induced sleep in *Drosophila* upregulates different nAchRα subunits is consistent with new findings that these subunits regulate appetitive memories in flies (*Pribbenow et al., 2022*) and that cholinergic systems more generally underpin learning and memory in this animal (*Barnstedt et al., 2016*). Yet learning and memory in flies clearly also benefit from quiet sleep, as evidenced by multiple studies using THIP as a sleep-promoting agent (*Dissel et al., 2015*; *Berry et al., 2015*; *Melnattur et al., 2021*). One view consistent with our findings and previous studies is that both kinds of sleep are crucial for optimal behavior: quiet sleep for cellular homeostasis and active sleep for circuit-level homeostasis. Manipulating these separately, alongside the non-overlapping pathways engaged by either kind of sleep, should help further disambiguate the functions potentially associated with these distinct sleep stages.

## Materials and methods

### Animals

*D. melanogaster* flies were reared in vials (groups of 20 flies/vial) on standard yeast-sugar-agar based media (1.0–1.5–0.5 g ratio) under a 12:12 light/dark (8 AM:8 PM) cycle and maintained at 25°C with 50% humidity. Adult, 3–5-day-old female, flies were used for all experiments and randomly assigned to experimental groups. Fly lines used for behavioral and RNA-sequencing experiments include R23E10-Gal4 (attp2; Bloomington 49032; Bloomington Drosophila Stock Center, Bloomington, IN) and UAS-CsChrimson-mVenus (attp18; Bloomington 55134; Provided by Janelia Research Campus, Ashburn, VA) (*Klapoetke et al., 2014*). For all two-photon experiments, flies with the genotype 10XUAS-Chrimson88-tdTomato (attp18)/+:LexAop-nlsGCaMP6f (VIE-260b; kindly provided by Barry J. Dickson)/+:Nsyb-LexA (attP2) (*Pfeiffer et al., 2012*), LexAop-PAGFP (VK00005)/R23E10-Gal4 were used. Optogenetically manipulated fly lines were maintained on food containing 0.5 mg/ml ATR (Merck, Darmstadt, Germany) for 24 hr prior to assay to allow for sufficient consumption. Pharmacologically manipulated flies were maintained on food with 0.1 mg/ml of gaboxadol (THIP) for the duration of behavioral experiment (*Dissel et al., 2015*).

### Two-photon imaging

Two-photon imaging was performed as described previously (*Tainton-Heap et al., 2021*) using a ThorLabs Bergamo series 2 multiphoton microscope. Fluorescence was detected with a High Sensitivity GaAsP photomultiplier tube (ThorLabs, PMT2000). GCaMP fluorescence was filtered through the microscope with a 594 dichroic beam splitter and a 525/25 nm band-pass filter.

For imaging experiments, flies were secured to a custom-built holder (*Tainton-Heap et al., 2021*). Extracellular fluid (ECF) containing 103 NaCl, 10.5 trehalose, 10 glucose, 26 NaHCO$_3$, 5 C$_6$H$_{15}$NO$_6$S, 5 MgCl$_2$ (hexa-hydrate), 2 sucrose, 3 KCl, 1.5 CaCl (dihydrate), and 1 NaH$_2$PO$_4$ (in mM) at room temperature was used to fill a chamber over the head of the fly. The brain was accessed by removing the cuticle of the fly with forceps, and the perineural sheath was removed with a microlance. Flies were allowed to recover from this for 1 hr before commencement of experiments. Imaging was performed across 18 z-slices, separated by 6 μm, with two additional flyback frames. The entire nlsGCaMP6f signal was located within a 256 × 256 px area, corresponding to 667 × 667 μm. Fly behavior was recorded with a Firefly MV 0.3MP camera (FMVU-03MTM-CS, FLIR Systems), which was mounted to a 75 mm optical lens and an infrared filter. Camera illumination was provided by a custom-built

infrared array consisting of 24 3 mm infrared diodes. Behavioral data was collected for the duration of all experiments.

For THIP experiments, an initial 5 min of baseline activity was captured, followed by perfusion of 0.2 mg/ml THIP in ECF onto the brain at a rate of 1.25 ml/min for 5 min. An additional 20 min of both brain and behavioral activity were recorded to allow visualization of the fly falling asleep on the ball as a result of THIP exposure. All flies were removed from imaging after an experiment and confirmed to have awoken by visual inspection. A subset of THIP-exposed flies (n = 3) remained in the imaging setup to measure brain activity upon recovery.

## Behavioral responsiveness probing

For probing behavioral responsiveness in the brain imaging preparation, flies walking on an air-supported ball were subjected to a 50-ms-long, 10-psi air puff stimulus, which was generated using a custom-built apparatus and delivered through a 3-mm-diameter tube onto the front of the fly. Flies were subjected to 10 pre-THIP stimuli at a rate of 1 puff/min to characterize the baseline response rate. Flies were then perfused with 0.2 mg/ml THIP in ECF for 5 min, followed by continuous ECF perfusion for the remaining experimental time. Five minutes after the fly had fallen asleep on the ball, a further 20 air puff stimuli were delivered, at a rate of 1 puff/min. Behavioral responses to the air puff were noted as a 'yes' (1) or 'no' (0), which were characterized as the fly rapidly walking on the ball immediately following the air puff. For statistical analysis, the pre-THIP condition was compared to either the first or last 10 min of the post-THIP condition.

## Imaging analysis

Preprocessing of images was carried out using custom-written MATLAB scripts and ImageJ.

Motion artifacts of the images were corrected as described previously (*Tainton-Heap et al., 2021*). Image registration was achieved using efficient sub-pixel image registration by cross-correlation. Each z-slice in a volume (18 z-slices and 2 flyback slices) is acquired at a slightly different timepoint compared to the rest of the slices. Hence, to perform volume (x,y,z) analysis of images, all the slices within a volume need to be adjusted for timing differences. This was achieved by using the ninth z-slice as the reference slice, and temporal interpolation was performed for all the other z-slices using 'sinc' interpolation. The timing correction approach implemented here is conceptually similar to the methods using fMRI for slice timing correction.

For each individual z-slice, a standard deviation projection of the entire time series was used for watershed segmentation with the 'Morphological segmentation' ImageJ plugin (*Legland et al., 2016*). Using a custom-written MATLAB (MathWorks) code, the mean fluorescent values of all pixels within a given ROI were extracted for the entire time series, resulting in a $n \times t$ array for each slice of each experiment, where $n$ refers to the number of neurons in each Z-slice, and $t$ refers to the length of the experiment in time frames. These grayscale values were z-scored for each neuron, and the z-scored data was transformed into a binary matrix where a value of >3 standard deviations of the mean was allocated a '1' and every value <3 standard deviations was allocated a '0.' To determine whether a neuron fired during the entire time series, a rolling sum of the binary matrix was performed, where 10 consecutive time frames were summed together. If the value of any of these summing events was greater than or equal to 7 (indicating a fluorescent change of >3 standard deviations in 7/10 time frames), a neuron was deemed to be active. For THIP sleep experiments, the 5 min of inactivity occurring after an initial 30 s of behavioral inactivity were used. After identifying firing neurons for each condition (wake vs. sleep), the percentage of active neurons was calculated in each slice by taking the number of active neurons and dividing it by the total number of neurons.

Traces of active neurons were used to calculate the number of firing events. This was done using the 'findpeaks' MATLAB function on the z-scored fluorescent traces, with the parameters 'minpeakheight' of 3 and 'minpeakdistance' of 30. Data resulting from this was crosschecked by taking the binary matrices of the time traces and finding the number of times each neuron met the activity threshold described above. Graph-theory analyses of neural connectivity were performed as described previously (*Tainton-Heap et al., 2021*).

## Behavioral sleep analysis

Behavioral data for flies in imaging experiments was analyzed as previously (*Tainton-Heap et al., 2021*) using a custom-written MATLAB code that measured the pixel change occurring over the legs of the fly on the ball over the entire time series. Data was analyzed and graphed using GraphPad Prism. All data were checked for Gaussian distribution using a D'Agostino–Pearson normality test prior to statistical testing. Data from THIP experiments was analyzed using a non-parametric Mann–Whitney test.

Sleep behavior in freely walking flies was analyzed with the *Drosophila* ARousal Tracking system (DART) as previously described (*Faville et al., 2015*). Prior to analysis, 3–5-day-old females were collected and loaded individually into 65-mm glass tubes (Trikinetics) that were plugged at one end with our standard fly food (see above), containing either 0.1 mg/ml THIP or 0.5 mg/ml ATR. Controls were placed onto normal food and housed under identical conditions as the experimental groups. The tubes were placed onto platforms (6 total platforms, 17 tubes per platform, up to 102 flies total) for filming. THIP-fed flies were monitored for 3 days/nights, and sleep data were averaged as no significant differences were detected across successive days. Those optogenetically activated were tracked for baseline day without red light, after which they were exposed to ultra-bright red LED (617 nm Luxeon Rebel LED, Luxeon Star LEDs, ON, Canada), which produce 0.1–0.2 mW/mm$^2$ at a distance of 4–5 cm with the aid of 723 concentrator optics (Polymer Optics 6° 15 mm Circular Beam Optic, Luxeon Star LEDs) for the duration of the experiment for optogenetic activation. Significance was determined by ANOVA with Tukey's multiple-comparisons test (GraphPad Prism). Sleep analysis in nAchRα knockout animals was performed using Trikinetics beam-crossing devices, with regular (>5 min) and short sleep (1–5 min) calculated as previously (*Tainton-Heap et al., 2021*).

## Sleep deprivation

Flies were sleep-deprived (SD) with the use of the previously described Sleep Nullifying Apparatus (SNAP), an automated SD apparatus that has been found to keep flies awake without nonspecifically activating stress responses (*Seugnet et al., 2009*). Vials containing no greater than 20 flies, which contained either standard food medium or medium containing 0.1 mg/ml THIP, were placed on the SNAP apparatus for continuous SD. The SNAP apparatus was programmed to snap the flies once every 20 s for the duration of the SD protocol.

## RNA-sequencing

Flies collected for RNA-sequencing analysis were first housed in vials containing either 0.5 mg/ml ATR or 0.1 mg/ml THIP for sleep induction, along with their genetically identical controls on standard food medium. Flies undergoing sleep induction by optogenetic activation with ATR and their controls were placed under constant red light from 8 AM until 6 PM to coincide with normal 12:12 light/dark cycles. Flies were collected after 1 hr (ZT 1) and 10 hr (ZT 10) post induction for immediate brain dissection and RNA extraction. For analysis of pharmacological sleep induction, flies were placed on THIP or normal food medium at 8 AM (ZT 0) and collected for dissection at 6 PM (ZT 10).

Whole fly brains were dissected in ice-cold RNA*later* (Sigma-Aldrich) with 0.1% PBST as per previously published protocol (*Chen et al., 2015*). The dissected brains were immediately pooled into five 1.5-ml Eppendorf tubes containing five brains (n = 25) each. Total RNA was immediately purified using TRIzol according to the manufacturer's protocols (Sigma-Aldrich) and stored at –80°C until the commencement of RNA-sequencing.

cDNA libraries were prepared using the Illumina TruSeq stranded mRNA library prep kit. Image processing and sequence data extraction were performed using the standard Illumina Genome Analyzer software and CASAVA (version 1.8.2) software. Cutadapt (version 1.8.1) was used to cut the adaptor sequences as well as low-quality nucleotides at both ends. When a processed read is shorter than 36 bp, the read was discarded by cutadapt, with the parameter setting of '-q 20,20 --minimum-length=36.' Processed reads were aligned to the *D. melanogaster* reference genome (dm6) using HISAT2 (version 2.0.5) (*Kim et al., 2015*), with the parameter setting of '--no-unal --fr --rna-strandness RF --known-splicesite-infile dm6_splicesites.txt.' This setting is to (i) suppress SAM records for reads that failed to align ('--no-unal'), (ii) specify the Illumina's paired-end sequencing assay and the strand-specific information ('--fr --rna-strandness RF'), and (iii) provide a list of known splice sites in *D. melanogaster* ('--known-splicesite-infile dm6_splicesites.txt'). Samtools (version 1.3) (*Li et al., 2009*) was

then used to convert 'SAM' files to 'BAM' files, sort and index the 'BAM' files. The 'htseq-count' module in the HTSeq package (v0.7.1) was used to quantitate the gene expression level by generating a raw count table for each sample (i.e., counting reads in gene features for each sample). Based on these raw count tables, edgeR (version 3.16.5) (*Robinson et al., 2010*) was adopted to perform the differential expression analysis between treatment groups and controls. EdgeR used a trimmed mean of M-values to compute scale factors for library size normalization (*Robinson and Oshlack, 2010*). It used the Cox–Reid profile-adjusted likelihood method to estimate dispersions (*McCarthy et al., 2012*) and the quasi-likelihood *F*-test to determine differential expression (*Lun et al., 2016*). Lowly expressed genes in both groups (the mean CPM < 5 in both groups) were removed. Differentially expressed genes were identified using the following criteria: (i) FDR < 0.05 and (ii) fold changes >1.5 (or logfc > 0.58). Gene Ontology enrichment analysis for differentially expressed genes was performed using the functional annotation tool in DAVID Bioinformatics Resources (version 6.8) (*Huang et al., 2009a*; *Huang et al., 2009b*).

## Gene expression

### RNA and cDNA synthesis

A quantitative reverse transcriptase PCR assay was used to confirm expression of genes enriched during THIP sleep induction. Nineteen candidate genes were selected (8 negatively and 11 positively) for the gaboxadol (THIP) sleep analysis and six genes (4 negatively and 2 positively) for the dFSB activation experiments. Total RNA was isolated using the Directzol RNA kit (ZymoResearch) from 20 adult brains per condition and each condition was collected in triplicate (i.e., three biological replicates). RNA quality was confirmed using a microvolume spectrophotometer NanoDrop 2000 (Thermo, USA) with only those resulting samples meeting optimal density ratios between 1.8 and 2.1 used. Up to 1 μg of total RNA was reverse transcribed using a High-Capacity cDNA Reverse Transcription Kit (Thermo) as per the manufacturer's protocols. The synthesis of cDNA and subsequent amplification was performed in max volumes of 20 μl per reaction using the T100 Thermal Cycler (Bio-Rad, USA). Thermocycle conditions were 25°C for 10 min, 37°C for 120 min, 85°C for 5 min, and held at 4°C. All cDNA was subsequently stored at −20°C until used. Target genes for THIP experiments included Pxt (CG7660, FBgn0261987), RpS5b (CG7014, FBgn0038277), Dhd (CG4193, FBgn0011761), CG9377 (CG9377, FBgn0032507), aKHr (CG11325, FBgn0025595), Acox57D-d (CG9709, FBgn0034629), FASN1 (CG3523, FBgn0283427), Pen (CG4799, FBgn0287720), CG10513 (CG10513, FBgn0039311), Gasp (CG10287, FBgn0026077), Act57B (CG10067, FBgn0000044), Bin1 (CG6046, FBgn0024491), verm (CG8756, FBgn0261341), CG16885 (CG16885, FBgn0032538), CG16884 (CG16884, FBgn0028544), CG5999 (CG5999, FBgn0038083), Fbp1 (CG17285, FBgn0000639), CG5724 (CG5724, FBgn0038082), and Eh (CG5400, FBgn0000564). Target genes for dFSB experiments included Vmat (CG33528, FBgn0260964), Dop1R1 (CG9652, FBgn0011582), Salt (CG2196, FBgn0039872), Dysb (CG6856, FBgn0036819), Irk3 (CG10369, FBgn0032706), Blos1 (CG30077, FBgn0050077). Housekeeping genes included Rpl32 (CG7939, FBgn0002626), Gapdh2 (CG8893, FBgn0001092), and Actin 5C (CG4027, FBgn0000042). Primer sequences can be found in *Supplementary file 3*.

### Quantitative real-time PCR

Quantitative (q) RT-PCR was carried out using the Luna Universal qPCR Master Mix (NEB) in the CFX384 Real-Time system (Bio-Rad). Cycling conditions were (1) 95°C for 60 s, (2) 95°C for 15 s, and (3) 60°C for 60 s with 39 cycles of steps 2 and 3. Melt curve analysis was then performed with the following conditions: (1) 95°C for 15 s, (2) 60°C for 60 s, and (3) 95°C for 15 s. Three biological replicates for each condition as well as three technical replicates per biological sample were loaded. Each experiment was then repeated on three separate occasions. Cq values and standard curves were generated using Bio-Rad CFX Manager Software to ensure amplification specificity. Results were normalized to the above housekeeping genes, and gene expression was calculated following the $2^{-\Delta\Delta Cq}$ method (Livak and Schmittgen 2001). All primer sequences used for these validation experiments are listed in *Supplementary file 3*.

### Gene knockouts and knockdowns

Dα1KO harbored an ends-out-mediated deletion of Dα1 in a w[1118] background with the X chromosome replaced with one from the wild-type line DGRP line 59 (*Somers et al., 2017*). For Dα2KO, 7 kb

of the Da2 locus was removed using the endogenously expressing Actin-Cas9 strain (BDSC 5490) crossed to a strain expressing two Dα2 gene locus-specific sgRNAs (Dα_sgRNA_start_1 5′CATGTTTA GCGCTGCAATGC, Dα2_sgRNA_end_1 5′TTACAAGCCATCTGCCTAG). Transgenic flies were generated using the pCFD4-U6:1_U6:3 tandem gRNAs construct. This was a gift from Simon Bullock. For Dα3KO (*Dα3Δ1020*), Dα4KO (*Dα4ΔBA*), Dα6KO, Dα7KO (*Dα7Δ6*), two sgRNAs were designed to target the start and the end of the coding sequence and cloned into either pU6-BbsI-gRNA or pCFD4 plasmids. These plasmids were then microinjected into *Drosophila* embryos to generate transgenic strains stably expressing sgRNAs. These strains were crossed to another strain expressing Cas9 under Actin promoter (ActinCas9). Their offspring were screened for deletion events with PCR and crossed to appropriate balancer strains to isolate and generate homozygous knockout strains. Full deletions were identified for all these subunit genes except for Dα3, which has two partial deletions at the 3′ and 5′ ends; these were verified to be true knockouts by the lack of RNA expression (*Perry et al., 2021*). ActinCas9 strain was used as genetic control for Dα3KO and Dα7KO, while this same strain with the X chromosome replaced with one from w$^{1118}$ (w$^{1118}$ActinCas9) was used as genetic control for Dα2KO, Dα4KO, and Dα6KO. The RNAi strain for the adipokinetic hormone receptor gene knockdown experiments (UAS-*AkhR*-RNAi) was obtained from the VDRC (KK109300). This RNAi construct was expressed in neurons across the fly brain by crossing the UAS flies to R57C10-Gal4 (*Jenett et al., 2012*). RNAi strains for the nicotinic alpha receptor gene knockdown experiments (UAS-nAChRα1-7-RNAi) were obtained from the BDRC. These were:

> nAChRα1 RNAi – BL# 28688 - y[1] v[1]; P{y[+t7.7] v[+t1.8]=TRiP.JF03103}attP2,
> nAChRα2 RNAi – BL# 27493 - y[1] v[1]; P{y[+t7.7] v[+t1.8]=TRiP.JF02643}attP2,
> nAChRα3 RNAi – BL# 27671 – y[1] v[1]; P{y[+t7.7] v[+t1.8]=TRiP.JF02750}attP2,
> nAChRα4 RNAi – BL# 31985 - y[1] v[1]; P{y[+t7.7] v[+t1.8]=TRiP.JF03419}attP2,
> nAChRα5 RNAi – BL# 77418 - y[1] sc[*] v[1] sev[21]; P{y[+t7.7] v[+t1.8]=TRiP.HMC06550}attP2,
> nAChRα6 RNAi – BL# 52885 - y[1] sc[*] v[1] sev[21]; P{y[+t7.7] v[+t1.8]=TRiP.HMC03623}attP40,
> nAChRα7 RNAi – BL# 27251 - y[1] v[1]; P{y[+t7.7] v[+t1.8]=TRiP.JF02570}attP2.

These RNAi constructs were expressed in the sleep-promoting neurons by crossing the UAS flies to R23E10-Gal4.

## Code, data, and materials availability

The data and analysis tools underpinning this study (brain imaging as well as RNA sequencing) are available online via Dryad: https://doi.org/10.5061/dryad.1rn8pk10x. See *Supplementary file 4* for details on accessing these datasets and code. Genetic reagents used in this study are available from the lead contact upon request.

## Acknowledgements

This work was supported by the National Health and Medical Research Council grant GNT1164499 to BvS and NIH R01 grant NS076980 to PJS and BvS.

## Additional information

### Funding

| Funder | Grant reference number | Author |
| --- | --- | --- |
| National Health and Medical Research Council | GNT1164879 | Bruno van Swinderen |
| National Institutes of Health | RO1NS076980 | Paul J Shaw |

The funders had no role in study design, data collection and interpretation, or the decision to submit the work for publication.

## Author contributions
Niki Anthoney, Data curation, Investigation, Visualization, Methodology, Writing – review and editing; Lucy Tainton-Heap, Software, Formal analysis, Validation, Investigation, Visualization, Methodology; Hang Luong, Eleni Notaras, Formal analysis, Investigation, Methodology; Amber B Kewin, Investigation, Methodology; Qiongyi Zhao, Data curation, Software, Formal analysis, Investigation, Methodology; Trent Perry, Philip Batterham, Resources, Supervision, Methodology; Paul J Shaw, Conceptualization, Resources, Funding acquisition, Investigation, Methodology, Project administration, Writing – review and editing; Bruno van Swinderen, Conceptualization, Resources, Data curation, Formal analysis, Supervision, Funding acquisition, Investigation, Visualization, Methodology, Writing – original draft, Project administration, Writing – review and editing

## Author ORCIDs
Trent Perry ⓘ http://orcid.org/0000-0002-8045-0487
Bruno van Swinderen ⓘ http://orcid.org/0000-0001-6552-7418

Joint Public Review: https://doi.org/10.7554/eLife.88198.3.sa1
Author Response https://doi.org/10.7554/eLife.88198.3.sa2

---

# Additional files

## Supplementary files
• Supplementary file 1. A comparison of sleep duration profiles (min/hr) during optogenetic and THIP-induced sleep. Tested with two-way ANOVA with Tukey's multiple-comparison test.

• Supplementary file 2. Raw data and statistics for RT-qPCR experiments.

• Supplementary file 3. Primer list for RT-qPCR validation experiments.

• Supplementary file 4. Readme file explaining how to access various components in the datasets made available.

## Data availability
All data generated or analysed during this study are included in the manuscript and supporting files. The data and analysis tools underpinning this study (brain imaging as well as RNA sequencing) are available online via Dryad: https://doi.org/10.5061/dryad.1rn8pk10x.

The following dataset was generated:

| Author(s) | Year | Dataset title | Dataset URL | Database and Identifier |
|---|---|---|---|---|
| Niki A, Lucy T-H, Hang L, Eleni N, Amber K, Qiongyi Z, Trent P, Philip B, Paul S, Bruno VS | 2023 | Data from: Experimentally induced active and quiet sleep engage non-overlapping transcriptional programs in *Drosophila* | https://doi.org/10.5061/dryad.1rn8pk10x | Dryad Digital Repository, 10.5061/dryad.1rn8pk10x |

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
