## [Editor Report · eLife assessment]

*Drosophila* is a powerful model organism for understanding the molecular and neural regulation of sleep. However, methodological limitations exist that would appear to limit the relevance of work done in the fly to our understanding of mammalian sleep. In this **important** work, the authors provide physiological, behavioral, and molecular evidence for the existence of two potential sleep stages in *Drosophila*. The experiments are generally well conducted and the authors' interpretations of their results are **solid** overall. Although technically innovative and conceptually provocative, there are aspects of the approaches used and results obtained that leave the central conclusions open to interpretation.

---

## [Referee Report · Joint Public Review]

Anthoney et al. provide an honorable attempt at furthering our understanding of the different sleep stages that may exist in *Drosophila*. The establishment of definable sleep stages/state in the fly model should be seen as a central goal in the field and this study represents an important step toward that goal. In particular, managing to draw parallels between sleep stages in flies and humans would make it relevant to use the power of fly genetics to better understand the molecular and cellular basis of these sleep stages. The authors use behavioral, physiological and transcriptomics approaches to describe the differences that exist between sleep triggered by optogenetic stimulation of dFB neurons and sleep induced by consumption of the sleep-promoting drug Gaboxadol (THIP). While there are still concerns regarding the interpretation of the major results, the authors have, in general, adequately responded to the reviewers' concerns.

The strengths of this work are:

1- The article is easy to read, and the figures are mostly informative.

2- The authors employ state-of-the-art techniques to measure neuronal activity and locomotion in a single assay.

3- The analysis of transcriptomic data is appropriate.

4- The authors identify many new genes regulated in response to specific methods for sleep induction. These are all potentially interesting candidates for further studies investigating the molecular basis of sleep. It would be interesting to know which of these genes are already known to display circadian expression patterns.

Concerns:

1- The fact that flies with dFB activation seem to keep a basal level of locomotor activity whereas THIP-treated ones don't is quite striking. Is it possible that there is an error with Figure 4C-D? Based on this, it is hard to believe that dFB stimulation and THIP consumption have similar behavioral effects on sleep.

2- The authors seem satisfied with the 'good correspondence' between their RNA-seq and qPCR results, this is true for only ~9/19 genes in Fig 6G and 2/6 genes in Fig 7G. The variability between the three biological replicates is not represented in the figure.

Comments

1- The fact that THIP-induced sleep persists long after THIP removal (Fig 3D) is intriguing. This suggests that the drug might trigger a sleep-inducing pathway that, once activated, can continue on its own without the drug.

2- The claim that induction of the two forms (active/quiet) of sleep produce distinct transcriptomic and physiological effects while producing highly similar behavioral effects makes it difficult to understand the relationships between the former changes with sleep state. In their response the authors argue that sleep behavior, as currently measured using duration of inactivity, is not necessarily expected to allow for a differentiation between active and quiet sleep. This further argues for a need for better physiological/molecular correlates of sleep state in the fly (a laudable goal of this very study). However, until clear behavioral correlates can be strongly associated with physiological/molecular correlates, we will be limited to speculation about this important issue.

3- The authors suggest that the duration of the periods of inactivity may not be particularly useful for defining sleep states/stages in the fly. If this is the case, it is certainly an important issue as this measurement is the behavioral criteria for defining sleep in the field. In this regard, it raises the question of the relationship between the active and quiet sleep states examined in this study and the growing evidence for a deep sleep state (characterized by, among other things, a lowered metabolism).

4- Although the methodological concerns regarding the dose of Gaboxadol and the controls for the optogenetic/transcriptomic experiments remain, the authors have explained the rationale for the experimental design they used.

5- Overall, the authors have managed to clearly illustrate the differences between dFB stimulation and THIP consumption on behavior, neuronal activity, and gene expression. In this regard, they have achieved what they claim in the title of the article. Overall, the results support the conclusions, however the main point to consider is that the methods employed here are artificial, and there is no guarantee at this stage that 'spontaneous' sleep has the same effects on the transcriptome than what is presented here.

5- This article represents an interesting attempt at addressing very important questions, and some of the data presented here, especially the RNA-seq, can be very useful for others. However, because of the lack of definitive conclusion about whether the results presented are applicable to 'natural' sleep, the impact of this article may remain limited beyond a relatively small field.

Comments to authors

The authors have suitably addressed all comments from this section.

---

## [Author Response]

The following is the authors’ response to the original reviews.

This important study shows that two methods of sleep induction in the fly, optogenetically activation of the dorsal fan-shaped body (which is rapidly reversible and maintains a neuronal activity signature similar to wakefulness), and Gaboxadol-induced sleep (which shuts down neuronal activity), produce distinct forms of sleep and have different effects on brain-wide neural activity. The majority of the conclusions of the paper are supported by compelling data, but the evidence supporting the claim that the two interventions trigger distinct transcriptional responses is incomplete.

Thank you for the helpful and detailed reviews. We feel that these have improved the manuscript considerably, and hopefully the additional figures in this Reply letter will help further convince our readers.

Public ReviewIn this study, Anthoney and coworkers continue an important, unique, and technologically innovative line of inquiry from the van Swinderen lab aimed at furthering our understanding of the different sleep stages that may exist in *Drosophila*. Here, they compare the physiological and transcriptional hallmarks of sleep that have been induced by two distinct means, a pharmacological block of GABA signaling and optogenetic activation of dorsal fan-shaped-body neurons. They first employ an incredibly impressive fly-on-the-ball 2-photon functional imaging setup to monitor neural activity during these interventions, and then perform bulk RNA sequencing of fly brains at different stages. These transcriptomic analyses leads them to (a) knocking out nicotinic acetyl-choline receptor subunits and (b) knocking down AkhR throughout the fly brain testing the impact of these genetic interventions on sleep behaviors in flies. Based on this work, the authors present evidence that optogenetically and pharmacologically induced sleep produces highly distinct brain-wide effects on physiology and transcription.The study is of significant interest, is easy to read, and the figures are mostly informative. However there are features of the experimental design and the interpretation of results that diminish enthusiasm.a- Conditions under which sleep is induced for behavioral vs neural and transcriptional studies1- There is a major conceptual concern regarding the relationships between the physiological and transcriptomic effects of optogenetic and pharmacological sleep promotion, and the effects that these manipulations have on sleep behavior. The authors show that these two means of sleep-induction produce remarkably distinct physiological and transcriptional responses, however, they also show that they produce highly similar effects on sleep behavior, causing an increase in sleep through increases in the duration of sleep bouts. If dFB neurons were promoting active sleep, the sleep it produces should be more fragmented than the sleep induced by the drug, because the latter is supposed to produce quiet sleep. Yet both manipulations seem to be biasing behavior toward quiet sleep.

This is a correct observation, which is already evident in our sleep architecture data (Figure 2E-H): chronic optogenetic sleep induction promotes longer sleep bouts that are similar in structure (bout number vs bout duration) to those produced by THIP feeding. Since our plots in Figure 2E-H follow the 5min sleep criterion cutoff, upon the Reviewer’s advice we re-analyzed our optogenetic experiments for short (1-5min) sleep. These are graphed below in Author response image 1. As can be seen, and as suspected by the Reviewer, the optogenetic manipulation does not increase the total amount of short sleep; indeed, it decreases it compared to baseline (these are for the exact same data as in Figure 2). Optogenetic sleep induction does not create a bunch of short sleep bouts.

**Author response image 1. sa2fig1:** Short sleep in optogenetic experiments. A. Average baseline ( ± SEM) 1-5min sleep across a day and night. B. Average ( ± SEM) 1-5min sleep in optogenenetically-activated flies, across a day and night.

We agree with the reviewer that this observation might seem inconsistent with the idea that optogenetic activation promotes active sleep, and that short sleep is active sleep. However, it does not necessarily follow that optogenetic activation has to produce short sleep. Indeed, we know from our brain imaging data (and the associated behavioral analysis) that active sleep will persist for as long as we induce it with red light. While we have not induced it for longer than 15 minutes (Tainton-Heap et al, Current Biology, 2021; Troup et al, J. of Neuroscience, 2023), this is already clearly longer than a <5min sleep bout. So our interpretation is that the longer sleep bouts induced by optogenetic activation are prolonged active sleep, rather than quiet sleep. In other words, this artificial sleep manipulation induces prolonged active sleep, rather than many short sleep bouts. This is of course different than what happens during spontaneous sleep. We have tried to be clearer about sleep bout durations in the revised manuscript (e.g., the new Figure 3), and we now admit early in the results (lines 376-380) that that we don’t know what optogenetic activation looks like in the fly brain beyond 15 minutes.

2- The authors show that the pharmacological block of GABA signaling and the optogenetic activation of dorsal fan-shaped-body neurons cause different responses on brain activity. Based on these recordings and the behavioral and brain transcriptomic data they then claim that these responses correspond to different sleep states and are associated with the expression and repression of a different constellation of genes. Nevertheless, neural activity in animals was recorded following short stimulations whereas behavioral and transcriptomic data were obtained following chronic stimulation. In this regard, it would be interesting to determine how the 12-hour pharmacological intervention they employed for their transcriptomic analysis changes neural activity throughout the brain - 12 hours will likely be too long for the open-cuticle preps, but an in-between time-point (e.g. 1h) would probably be equally informative.

The longest we’ve imaged brain activity for optogenetic sleep induction is 15 minutes, as discussed above. We see no changes in activity across this time, which would normally have led to a quiet sleep stage in spontaneous sleep recordings. Whole-brain imaging after 10 hours of optogenetic sleep induction (our RNA collection timepoint) is not realistic, and even 1 hour is difficult. We have however conducted overnight electrophysiological recordings (with multichannel silicon probes), where we activated the same R23E10 neurons for successive 20-minute bouts (alternating with 20min of no red light). We are preparing this work for publication (Van De Poll, et al). We see no evidence of optogenetic activation of this circuit ever producing anything resembling quiet sleep. Since we are not in a position to provide this new electrophysiological data in the current study, we are careful to clarify that we have not investigated what brain imaging looks like after chronic optogenetic activation (lines 376-380). We are showing through diverse lines of evidence that what is called sleep can look different in flies.

b- Efficiency of THIP treatment under different conditions1- There are no data to quantify how THIP alters food consumption. It is evident that flies consume it otherwise they would not show increased sleep. However, they may consume different amounts of food overall than the minus THIP controls. This might have an influence on the animal's metabolism, which could at least explain the fact that metabolism-related genes are regulated (Figure 5). Therefore, in the current state, it is not possible to be certain that gene regulation events measured in this experiment are solely due to THIP effects on sleep.

We have two arguments against this reasonable criticism. First, as discussed above, the optogenetic flies are sleeping at least as much as the THIP-fed flies, so in principle they also might be feeding less. But we see no metabolic gene downregulation in the optogenetic dataset. We include this counterargument in the discussion (lines 752-756). Then, together with our co-author Paul Shaw we have shown that THIP-fed flies are not eating less compared to controls (Dissel et al, Current Biology, 2015), by tracking dye consumption. We show those results again below in Author response image 2 to support our reasoning that feeding is not an issue.

**Author response image 2. sa2fig2:** Flies were fed blue dye in their food while being sleep deprived (SD), or while being induced to sleep with 0. 1mg/ml THIP in their food, or both. Dye consumption was measured in triplicate for pooled groups of 16 flies. Average absorbance at 625nm (±stan dev) is shown. Experiments were not significantly different (ANOVA of means).

2- A similar problem exists in the sleep deprivation experiments. If flies are snapped every 20 seconds, they may not have the freedom to consume appropriate amounts of food, and therefore their consumption of THIP or ATR may be smaller than in non-sleep deprived controls. Thus, it would be crucial to know whether the flies that are sleep-deprived (i.e. shaken every 20 seconds for 12 hours) actually consume comparable amounts of food (and therefore THIP) as those that are undisturbed. If not, then perhaps the transcriptional differences between the two groups are not sleep-specific, but instead reflect varying degrees of exposure to THIP.

Please see our response to the similar critique above, and how Figure R2 addresses this concern.

3- The authors should further discuss the slow action of THIP perfusion vs dFB activation, especially as flies only seem to fall asleep several minutes after THIP is being washed away. Is it a technical artifact? If not, it may not be unreasonable to hypothesize that THIP, at the concentration used, could prevent flies from falling asleep, and that its removal may lower the concentration to a point that allows its sleep-promoting action. The authors could easily test this by extending THIP treatment for another 4-5 minutes.

The reviewer is partially correct in suggesting a technical artifact: THIP does not get washed away immediately after 5min of perfusion. The drip system we employ means that THIP concentration will slowly increase to the maximum concentration of 0.2mg/ml, and then slowly get diluted away at a rate of 1.25ml/minute (this is all in the Methods). In a previous study (Yap et al, Nature Communications, 2017) we used this exact same perfusion procedure to test a range of THIP concentrations, and settled on 0.2mg/ml as the lowest that reliably induced quiet sleep within 5 minutes. Higher concentrations induced quiet sleep faster, so the alternate explanation proposed by the Reviewer is not supported. We feel that our previous electrophysiological study provided the necessary groundwork for using the same approach and dosage here for our whole-brain imaging readout.

c- Comments regarding the behavioral assays1- L319-322: the authors conclude that dFB stimulation and THIP consumption have similar behavioral effects on sleep. However, this is inaccurate as in Figure S1 they explain that one increases bout number in both day and night and the other one only during the day.

We have now added a caveat about night bout architecture being different (lines 353-356). Figure S1 is now Figure 3.

2- The behavioral definitions used for active and quiet sleep do not fit well with strong evidence that deep sleep (defined by lowered metabolic rates) is probably most closely associated with bouts of inactivity that are much longer than the >5min duration used here, i.e., probably 30min and longer (Stahl et al. 2017 Sleep 40: zsx084). Given that the authors are providing evidence that quiet sleep is correlated with changes in the expression of metabolism related genes, they should at least discuss the fact that reductions in metabolism have been shown to occur after relatively long bouts of inactivity and might reconsider their behavioral sleep analysis (i.e., their criteria for sleep state) with this in mind.

Interestingly, induced sleep bout durations are on average longer for the optogenetic manipulation (40min vs 25min); this was evident in Figure S1C vs S1F (now Figure 3). So as discussed above, this provides a counterargument for sleep bout duration alone being indicative of metabolic processes associated with quiet sleep: the optogenetic dataset did not uncover metabolic-related pathways as relevant to that sleep manipulation. We refer to Stahl et al, Sleep, 2017, in our discussion (lines 748-750), making exactly this point about metabolic rates being decreased in longer sleep bouts, and flowing up with our observation that optogenetic flies sleep just as much, and their bouts are actually longer. So clearly different processes must be involved.

d- Comments regarding the recordings of neuronal activity1- There is an additional concern regarding the proposed active and quiet sleep states that rest at the heart of this study. Here these two states in the fly are compared to the REM and NREM sleep states observed in mammals and the parallels between active fly sleep and REM and quiet fly sleep and NREM provide the framework for the study. The establishment of such parallel sleep states in the fly is highly significant and identifying the physiological and molecular correlates of distinct sleep stages in the fly is of critical importance to the field. However, the proposal that the dorsal fan shaped body (dFB) neurons promote active sleep runs counter to the prevailing model that these neurons act as a major site of sleep homeostasis. If quiet sleep were akin to NREM, wouldn't we expect the major site of sleep homeostasis in the brain to promote it? Furthermore, the authors state that the effects of dFB neuron excitation on transcription have "almost no overlap" (line 500) with the transcriptomic effects of sleep deprivation (Supplementary Table 3), which is not what would be expected if dFB neurons are tracking sleep pressure and promoting sleep, as suggested by a growing body of convergent work summarized on page four of the manuscript. Wouldn't the 10h excitation of the dFB neurons be predicted to mimic the effects of sleep deprivation if these neurons "...serve as the discharge circuit for the insect's sleep homeostat..." (line 60)? Shouldn't their prolonged excitation produce an artificial increase in sleep drive (even during sleep) that would favor deep, restorative sleep? How do the authors interpret their results with regard to the current prevailing model that dFB neurons act as a major site of sleep homeostasis? This study could be seen as evidence against it, but the authors do not discuss this in their Discussion.

These are all excellent and thoughtful points, which have made us re-think parts of our discussion. First off, the potential comparison with REM and NREM is entirely speculative, and we have tried to make that more obvious in introduction and the discussion (e.g, see lines 43, 708, 818). The evidence that the FB neurons (and maybe others) are involved in the homeostatic regulation of sleep is well-supported in the literature, so that part of the discussion holds. However, we concede that the timing of our sleep manipulations could benefit from more explanation. We conducted these during the flies’ subjective day, after the animals had presumably had a good night’s sleep. This means that we induced either kind of sleep for 10 daytime hours, which presumably replaced whatever behavioural states would ‘naturally’ be happening during the day. Female flies sleep less during the day than at night, and we have shown in previous work that daytime sleep quality is different than night-time sleep (van Alphen et al, Journal of Neuroscience, 2013), leading us to suggest that most ‘deep’ or quiet sleep happens at night, for flies. Following this reasoning, daytime optogenetic activation might not be depriving flies of much quiet sleep, or accumulating a deep sleep drive as the Reviewer proposes. Rather, both induced sleep manipulations could be providing 10 hours of either kind of sleep that the flies don’t really ‘need’. Why did we design it this way? Firstly, we were interested in simply asking what these chronic sleep manipulations do to gene expression in rested flies, and how they might be similar or different. We focussed on daytime manipulations to avoid precisely the confound of sleep pressure, and also because we observed red-light artifacts at night for our optogenetic experiments (which we reported). Our sleep deprivation strategy was designed specifically as a control for the THIP (Gaboxadol) experiments, to control for non-sleep related effects of the drug (see below our rationale for why this was less crucial for the optogenetic experiments). In conclusion, we had a logical rationale for how the experiments were done, centred on the straightforward question of whether these two different approaches to sleep induction were having similar effects in well-rested flies. In retrospect, we were not anticipating the Reviewer’s thoughtful logic regarding the dFB’s potential role in also regulating deep sleep homeostasis. We now provide some discussion along these lines to make readers aware of this line of reasoning, as well as our rationale for why prolonged optogenetic sleep induction was not sleep-depriving (lines 768-777).

2- Regarding the physiological effects of Gaboxadol, to what extent is the quieting induced by this drug reminiscent of physiology of the brains of flies spontaneously meeting the behavioral criterion for quiet sleep? Given the relatively high dose of the drug being delivered to the de-sheathed brain in the imaging experiments (at least when compared to the dose used in the fly food), one worries that the authors may be inducing a highly abnormal brain state that might bear very little resemblance to the deeply sleeping brain under normal conditions. As the authors acknowledge, it is difficult to compare these two situations. Comparing the physiological state of brains put to sleep by Gaboxadol and brains that have spontaneously entered a deep sleep state therefore seems critical.

As discussed above, our Gaboxadol (THIP) perfusion concentration (0.2mg/ml) was the minimal dosage that effectively induced sleep within 5 minutes, based upon previously published work (Yap et al, Nature Communications, 2017). Lower concentrations were unreliable, with some never inducing sleep at all. Comparisons with feeding THIP are tenuous, and we make that clear in our discussion (lines 731-735). Nevertheless, the Reviewer makes an excellent point about comparisons with spontaneous ‘quiet’ sleep. Here, we feel well supported (please see Author response image 3 below, comparing THIP-induced sleep (this work, B) and spontaneous sleep (A) from previous study). In our previous study (Tainton-Heap et al, 2021) we showed that neural activity and connectivity decreases during spontaneous quiet sleep. This is what we also see with THIP perfusion. In contrast, in Troup et al, J. of Neuroscience (2023) we confirm that neither neural activity nor connectivity changes during optogenetic R23E10 activation, and general anesthesia – unlike THIP – does NOT produce a quiet brain state. Our finding that THIP effects are nothing like general anesthesia (at the level of brain activity levels) suggests a physiological sleep state closer to spontaneous quiet sleep. We elaborate on this important observation in our results, also pointing to crucial differences with general anesthesia (lines 411-415).

**Author response image 3. sa2fig3:** THIP-induced sleep resembles quiet spontaneous sleep. A. Calcium imaging data from spontaneously sleeping flies, taken from Tainton-Heap et al, 2021. Left, percent neurons active; right, mean degree, a measure connectivity among active neurons. Both measures decrease during later stages of sleep. B. Calcium imaging data from flies induced to sleep with 5min of 0.2mg/ml THIP perfusion (this study). Left, percent neurons active; right, mean degree. Both measures are significantly decreased, resembling the later stages of spontaneous sleep, which we have termed ‘quiet sleep. Hence THIP-induced sleep resembles quiet sleep. Note that the genetic background is different in A and B, hence the different baseline activity levels.

3- There are some issues with Figure 3, in particular 3C-D. It is not clear whether these panels show representative traces or an average, however both the baseline activity and fluorescence are different between C and D, in particular in their amplitude. Therefore, it is difficult to attribute the differences between C and D to the stimulation itself or to the previously different baseline. In addition, the fact that flies with dFB activation seem to keep a basal level of locomotor activity whereas THIP-treated ones don't is quite striking, however it is not being discussed. Finally, the authors claim that the flies eventually wake up from THIP-induced sleep (L360-361), however there are no data to support this statement.

These are representative traces, which is a way of showing the raw calcium data (Cell ID) so readers can see for themselves that one manipulation silences whereas the other does not – even though flies become inactive for both. The Y-axis scale is standard deviation of the experiment mean. Since THIP decreases neural activity, then the baseline is comparatively higher. Since optogenetic activation does not change average neural activity levels, the baseline is centered on zero. This is an outcome of our analysis method and does not reflect any ‘true’ baseline. We have now clarified this in our figure legend. We now also confess that flies rendered asleep optogenetically can be ‘twitchy’ (line 374). Finally, we show data for 3 flies that were recorded until they woke up. The rest were verified behaviorally, after the experiment. This is now explained in the Methods.

4- In Figure 4C, it is strange that the SEM is always exactly the same across the whole experiment. Readers should be aware that there might have been an issue when plotting the figure.

This is not a mistake, the standard errors are just all quite close (between 0.17 and 0.22). This is because of the way we did the analysis, asking how many flies responded to each stimulus event, with incremental levels of responsiveness. This is explained in the Methods. The figure makes the important point of sleep and recovery.

e- Comments regarding the transcript analyses1- General comment: the title of this manuscript is inaccurate - the "transcriptome" commonly refers to the entirety of all transcripts in a cell/tissue/organ/animal (including genes that are not differentially expressed following their interventions), and it is therefore impossible to "engage two non-overlapping transcriptomes" in the same tissue. Perhaps the word "transcriptional programs" or transcriptional profiles" would be more accurate here?

We thank the Reviewer for this advice and have changed the title as proposed.

2- Given the sensitivity of transcriptomic methods, there is a significant concern that the optogenetic experiments are not as well controlled as they could be. Given the need for supplemental all-trans retinal (ATR) for functional light gating of channelrhodopsins in the fly, it is convenient to use flies with Gal4-driven opsin that have not been given supplemental ATR as a negative control, particularly as a control for the effects of light. However, there is another critical control to do here. Flies bearing the UAS-opsin responder element but lacking the GAL4 driver and that have been fed ATR are critical for confirming that the observed effects of optogenetic stimulation are indeed caused by the specific excitation of the targeted neurons and not due to leaky opsin expression, or the effect of ATR feeding under light stimulation or some combination of these factors. Given the sensitivity of transcriptomic methods, it would be good to see that the candidate transcripts identified by comparing ATR+ and ATR- R23E10GAL4/UAS-Chrimson flies are also apparent when comparing R23E10GAL4/UAS-Chrimson (ATR+) with UAS-Chrimson (ATR+) alone.

We have not done these experiments on UAS-Chrimson/+ controls. Like many others in our field, we viewed non-ATR flies as the best controls, because this involves identical genotypes. Since we were however aware that ATR feeding itself could be affect gene expression, we specifically checked for this with our early (1hour) collection timepoint. We only found 26 gene expression differences between ATR and -ATR flies at this early timepoint, compared with 277 for the 10-hour timepoint. We detail this rationale in our results, explaining why this is a convincing control for ATR feeding. If there was leaky opsin expression / activity, this would have been evident in our design. Regarding the cumulative effect of light, this would also have been accounted in our design, as only 1 hour would have elapsed in our first timepoint compared to 10 hours in our second. While the Reviewer is correct in saying that parental controls are called for in many *Drosophila* experiments, this becomes quickly unmanageable in transcriptomic studies, which is exactly why well-designed +ATR vs -ATR comparisons in the exact same strain are most appropriate. We feel that our 1-hr timepoint mostly addresses this concern.

3- Figures about qPCR experiments (5G and 6G) are problematic. First, whereas the authors seem satisfied with the 'good correspondence' between their RNA-seq and qPCR results, this is true for only ~9/19 genes in 5G and 2/6 genes in 6G. Whereas discrepancies are not rare between RNA-seq and qPCR, the text in L460-461 and 540-541 is misleading. In addition, it is unclear whether the n = 19 in L458 refers to the number of genes tested or the number of replicates. If the qPCR includes replicates, this should be more clearly mentioned, and error bars should be added to the corresponding figures.

We consider that our qPCR validations were convincing, as they were all mostly changed in the ‘right’ direction. We agree that are some discrepancies, so have modified our language to reflect this. We have also clarified that 19 refers to the number of genes validated by qPCR in that THIP dataset. All qPCRs involved three technical replicates. We prefer to keep these histograms the way they are to convey these simple trends. For complete transparency, we now provide a supplemental Excel worksheet with all of the qPCR data, alongside corresponding RNAseq data and stats for the selected genes (Supplementary Table 9).

4- There is a lack of error bars for all their RNAseq and qPCR comparisons, which is particularly surprising because the authors went to great lengths and analyzed an applaudably large amount of independent biological replicates, yet the variability observed in the corresponding molecular data is not reported.

The genes reported in each of our datasets and associated supplemental figures and tables were all significant, as determined by criteria outlined in the Methods. However, we appreciate that readers might want to get a sense of the values and variances involved, as well as access to the entire gene datasets. We now provide all of these as additional ‘sheets’ in our existing supplemental tables (S2-S7), so this should be very easy to navigate and evaluate. In addition to the previously provided lists for significant genes, in the second Excel sheet (‘All genes’) readers will be able to see the data for all 5 replicates, for the significant genes as well as all other ~15,000 genes (listed in alphabetical order). We feel that this will be a helpful resource, because admittedly significance thresholds can still be a little arbitrary and some readers might want to look up ‘their’ genes of interest.

Comments to authorsOther comments1- Text in L441 & 606 is misleading. According to ref 52, AkhR is involved specifically in starvation-induced sleep loss, and not in general sleep regulation.

Corrected.

2- The language used in L568-570 and 573-574 is confusing. The authors should specify that the knock down of cholinergic subunits, rather than the subunits themselves is what causes sleep to increase or decrease.

Corrected.

3- The authors' investigation of cholinergic receptor subunits function is very preliminary, and it is difficult to draw any conclusion from what is presented here. In particular, their behavioral data is difficult to reconcile with the RNA-seq data showing overexpression of both short sleep increasing and short sleep decreasing subunits. Without knowing where in the brain these subunits are required for controlling sleep, the data in Figure 7 is difficult to appreciate.

We have now conducted additional experiments where we specifically knocked down these alpha receptor subunits (all 7 of them) in the R23E10 neurons. This seemed an obvious knockdown location, to determine if any of these subunits regulated activity in the same sleep promoting neurons that were the focus of this study. We found that alpha1 knockdown in these neurons had similar sleep phenotypes, which we believe is an important result. Since this functional localisation is a logical ending for the paper, we have now made it the final figure.

Suggestions & comments1- It would be interesting if the authors could discuss their findings that metabolism genes are downregulated in THIP flies in the context of recent work that showed upregulation of mitochondrial ROS after sleep deprivation (Kempf et al, 2019).

We now add the Kempf 2019 reference and allude to how those findings could be consistent with ours.

2- The fact that THIP-induced sleep persists long after THIP removal (Fig 3D) is very intriguing and interesting. This suggests that the drug might trigger a sleep-inducing pathway that can continue on its own without the drug, once activated.

This is correct, and in stark contrast to the optogenetic manipulation we employ, which does not appear to show such sleep inertia. We have now added a sentence highlighting this interesting difference (lines 394-396).

3- The authors identify many new genes regulated in response to specific methods for sleep induction. These are all potentially interesting candidates for further studies investigating the molecular basis of sleep. It would be interesting to know which of these genes are already known to display circadian expression patterns.

By providing all of the gene lists, these are now available to ask questions such as these. We hesitate however to delve into this domain for this work, as our main goal was to compare these two kinds of sleep in flies.

4- The brain-wide monitoring of neural activity invites a number of very exciting follow-up experiments - most importantly, it would be fascinating to establish, which neurons are active in the different phases the authors describe! Are these neurons that are involved in transmitting external visual stimuli to the central brain? Do they also project into the central complex? They could make use of the large collection of existing driver lines in the fly and they could also exploit the extraordinary knowledge of the connectome and transcriptome of the fly brain.

Thank you for sharing our enthusiasm for these likely future directions.

5- The Dalpha2,3,4,6 and 7 Knock-out strains they generate will be a useful reagent for the *Drosophila* neuroscience community once the efficiency/success of the knock-out has been confirmed by qPCR.

These knockout strains have all been confirmed by our co-authors Hang Luong, Trent Perry, and Philip Batterham. These knockout confirmations are outlined in publications that we reference (Perry et al, 2021).

Materials and methods:1- This study has employed custom-built apparatus and custom-written code/scripts, but these do not appear to be available to the reader. For the sake of replicability, the authors should make these available.

The code/scripts are available via the University of Queensland research data management system as described in the Methods, and can be sent by the Lead Contact. The imaging hardware and analysis code are identical to what was described in a previous publication, and available as directed therein (Tainton-Heap et al, 2021).

2- Also, the authors should give details on the food used to rear their flies. Fly media comes in several common forms and sleep is sensitive to diet.

This has now been elaborated in the beginning of the Methods.

3- The light regime used for optogenetic excitation of dFB neurons consists of 12h of uninterrupted bright red LED light. Most optogenetic stimulations consist of pulsed high frequency flashes interlaced with pauses in illumination. Can dFB neurons be driven constitutively with 12 hours of bright light?

We showed in Tainton-Heap (2021) that 7Hz pulsed red light had exactly the same effect on R23E10/Chrimson readouts as continuous red light, which is why we opted here to provide continuous red light. That optogenetic sleep induction can be driven continuously for 12 hours is evident by our 24-hour sleep profiles. However, we agree that one could question whether sleep quality is similar after 12 hours. To address this, we did an additional experiment where we stimulated the flies hourly, to determine if their behavioural responsiveness to mechanical stimuli changed over the course of continued sleep induction, for both optogenetic and THIP-induced sleep. We present the data below in Author response image 4. As can be seen in these new analyses, while optogenetic sleep induction persists across 12 daytime hours (speed is close to zero throughout), flies do indeed become more responsive later in the day. This could have two different interpretations: either some sleep functions are being satisfied over time, or the activation regime is becoming less effective over time. Either way, these data show that at our 10-hour daytime timepoint, unstimulated flies are still largely inactive, even though their arousal thresholds might have gradually changed; so the uninterrupted red-light regime is still effective. The comparison with THIP is interesting: here there does not seem to be a change in responsiveness over time; the drug just decreases behavioral responsiveness throughout. Together, these experiments support our view that both approaches are sleep-promoting throughout the 12-hour day, although we appreciate that sleep quality is not identical.

**Author response image 4. sa2fig4:** Responsiveness to mechanical stimuli. (A) The average speed of baseline (grey) and optogenetically-activated flies (green) across 24 hours. Red dots indicate vibration stimulus times. (B) The average speed of control (grey) and THIP-fed flies (blue) across 24 hours. Flies are all R23E10/Chrimson. N = 87 for optogenetic, n = 88 for -THIP, n = 85 for +THIP.

4- The authors use the SNAP apparatus to prevent THIP-treated flies from sleeping to tease out possible sleep-independent effects. This is an excellent control. Why have the authors not done the same with the optogenetic treatment? It's surprising not to see this control given the concern the authors express (lines 501 - 502) that the dFB manipulation might be paralyzing awake flies, which certainly seems possible given the light regimes used. Why not test this directly with SNAP?

We appreciate that this may have been a valuable additional control. However, we designed this control for the THIP experiments specifically because of concerns about THIP’s (yet unknown) mechanism of action in flies. THIP is a gabaergic drug with most likely many off-target effects that have little to do with sleep, hence the need for a control where we compare to flies that ingested THIP but have been prevented from sleeping. In contrast, R23E10-driven sleep induction is exactly that, a circuit when activated that induces sleep. Whatever specific neurons might really be involved, the Gal4 circuit is sleep-inducing. This is well supported by multiple publications. The most appropriate control for assessing transcriptomic effects during optogenetic sleep here is not preventing sleep, but rather no increased sleep in flies that have not ingested ATR, and comparing that to effects of ATR alone, which is what we have done. Adding a sleep-deprivation layer onto both of these analyses may have been interesting, but a lot more analyses and not strictly required to identify relevant sleep-related genes. We have rephrased the misleading sentence about paralyzing flies, to instead clarify that lack of overlap with the SD dataset suggests that optogenetic activation is not preventing sleep functions from being engaged.

5- A pairwise comparison of ZT01 and ZT10 does not address circadian expression cycles in a meaningful way. There will be strong effects of the LD cycle here. I suggest toning this down. (Though it is gratifying to see the expected changes in the core clock genes.)

We have changed the language from ‘circadian’ to ‘light-dark’ to address this, although have kept the word ‘circadian’ when referring specifically to genes such as per, clock, timeless, etc.

6- Line 109: There is a reference missing.

We now provide the relevant reference.

Results1- General comment regarding the figures: a general effort could be made to improve the design and quality of the figures and make them more readable. There are a lot of issues such as stretched or misaligned text, badly drawn frames, etc.

We think we know which figures this might relate to (e.g., Figures 3,4B), so we have adjusted where appropriate.

2- Instead of 'dFB-induced' (e.g., L77) it would be more accurate to use 'optogenetically-induced'

Thank you for this helpful advice. We have changed our language throughout to say ‘optognetically-induced’

3- Figure S1 should be integrated in the main figure to make the quantification more easily 4accessible.

We have integrated Figure S1 into the main figures. It is now Figure 3.

5- It would be good to include red light controls in Figure 2C, E, G.

Making Figure S1 a main figure has better highlighted the fact that we have done red light controls (‘baseline’).

6- line 313: Fig2E-H - these graphs would benefit if the authors made it more obvious where the maximum sleep amount would fall - i.e. the combination of bouts and minutes that add up to 12 hours (and therefore the entire day/night)

If a fly were to sleep uninterrupted for all 12 hours of a day or night, that would amount to a sleep bout 720 minutes long. We do not feel that identifying this maximum on these graphs would be helpful. It should be clear from the data that a floor is reached with very few sleep bouts exceeding 60 minutes in our paradigm. To help orient the reader though, we now clarify in the figure legend that the maximum is 720 minutes or 12 hours.

7- Fig. 2B, D: It was not clear why the authors took the 3-day average here. Doesn't that lead to a whole range of very different behaviors? I could, perhaps naively, imagine that a fly's behavior changes after 2 days of almost-permanent sleep?

We took the 3-day average because the effect of THIP on each successive day was not significantly different (see Author response image 5, below). Flies wake up enough to have a good feed (see Author response image 2) and then go back to sleep. Since this is however an important point raised by the reviewer, we now mention in the Methods that sleep duration was not different among the 3 averaged days and nights (lines 193-195).

**Author response image 5. sa2fig5:** Data from THIP feeding experiment (Figure 2B) in manuscript, separated into 3 successive days and nights, with THIP-fed flies (blue) compared to controls (white). Averages ± SD are shown, samples sizes are the same as in Figure 2D. No THIP data was significantly different across days and nights (ANOVA of means).

8- In Figure 2C the authors compare optogenetically induced to "spontaneous sleep," which I think refers to baseline sleep before stimulation, according to the figure. I think the proper comparison would be to the red light control (ATR-); though see the comment above regarding optogenetic controls.

This information was provided in Figure S1. We now provide it as a main Figure 3, as requested above.

We also made a point about red light having an effect at night, which is why we focussed on daytime effects for our transcriptomic comparisons. We feel that the ATR-fed flies (minus red light) are an appropriate control here for optogenetically-induced sleep: same exact genotype and ATR feeding, just no optogenetic activation. We therefor would prefer to keep these graphs as they are, especially since we show -ATR data subsequently.

9- Figures 3A and 4A are redundant; Figure 3B has some active ROIs that are outside of the brain. I am not sure how this is possible?

We have removed the redundant 4A and replaced it with the THIP molecule to clearly signal what this figure is focussed on. In Figure 3B (now 4B), the brain mask is a visual estimate made from the middle of the image stack. Some neurons in other layers are outside this single-layer estimate. All neurons were all accounted for.

10- Figure 4B is confusing. It took me a while to understand and so it can do with re-drawing in a more accessible way.

We agree that this was confusing, e.g. there were too many arrows. We have redrawn and simplified (Now 5A).

11- The authors state that flies wake up from THIP-induced sleep on the ball, but in Figure 4D there appears to be fewer samples for flies who have woken up from THIP (3) compared to those observed before THIP administration. Are flies dying?

None of the flies died. Most flies were removed from imaging to confirm recovery, while 3 were left in our imaging setup to measure brain activity upon recovery. These results are in Figure 5C and now clarified in the Methods.

12- Fig5C,D: I'm surprised that by far the most significant changes (in terms of log2-FC and p-val) occur in the sleep-deprived flies? It is not clear to me what the authors mean by effects that "relate waking process"? Perhaps they could elaborate on this?

We have removed the phrase ‘relates to waking processes’. We now also remark on the high level of fold-change in many of these genes but refrain from discussing this further in the results. It is interesting though.

13- The sentence in L425-428 is unclear - it would be good to rephrase this.

We have rephrased this sentence, hopefully it’s clearer now.

14- Text in L544-545 is confusing. What do you mean by 'less clear'?

We have replaced ‘less clear’ with ‘not dominated by a single category’.

15- It is unclear what is the control in Fig 7A. It would be good to mention what strain was used.

Different knockout strains had different controls. These are identified in the figure legend and Methods.

16- L579-581: it would be helpful to include this data in a supplementary figure.

We now provide this as a supplementary figure as requested (Supplementary Figure 6).

17- There is no information about R57C10 in the methods - it would be good to explain which neurons this line labels, and why you chose it.

We now clarify in the methods that R57C10-Gal4 is a pan-neural driver, and provide a reference.

18- Table S5 - If I'm not mistaken then the first line should say 1h, not 10h.

Corrected